# Image-quality metric system for color filter array evaluation

**Tae Wuk Bae** [ID] *

Daegu-Gyeongbuk Research Center, Electronics and Telecommunications Research Institute, Daegu, South Korea

* nanninggo@gmail.com, twbae@etri.re.kr

**Data Availability Statement:** All relevant data are within the manuscript and its Supporting Information files.

**Funding:** This research received no external funding

## Abstract

A modern color filter array (CFA) output is rendered into the final output image using a demosaicing algorithm. During this process, the rendered image is affected by optical and carrier cross talk of the CFA pattern and demosaicing algorithm. Although many CFA patterns have been proposed thus far, an image-quality (IQ) evaluation system capable of comprehensively evaluating the IQ of each CFA pattern has yet to be developed, although IQ evaluation items using local characteristics or specific domain have been created. Hence, we present an IQ metric system to evaluate the IQ performance of CFA patterns. The proposed CFA evaluation system includes proposed metrics such as the moiré robustness using the experimentally determined moiré starting point (MSP) and achromatic reproduction (AR) error, as well as existing metrics such as color accuracy using CIELAB, a color reproduction error using spatial CIELAB, structural information using the structure similarity, the image contrast based on MTF50, structural and color distortion using the mean deviation similarity index (MDSI), and perceptual similarity using Haar wavelet-based perceptual similarity index (HaarPSI). Through our experiment, we confirmed that the proposed CFA evaluation system can assess the IQ for an existing CFA. Moreover, the proposed system can be used to design or evaluate new CFAs by automatically checking the individual performance for the metrics used.

## 1. Introduction

An improvement in the structure and operation, a technique for reducing the pixel size, and a wide dynamic range in CMOS image sensors have recently become important issues for the development of smaller advanced cameras. The most representative Bayer CFA is often used to implement a single sensor for color images [1]. Camera manufacturers are developing CFAs with different colors and structures to improve the picture quality. Factors evaluating the camera quality include the color accuracy, color difference, image contrast, and dynamic range etc.

The development of new image sensors, particularly CFAs, means the development of CFA elements and structures that have a better IQ in terms of the human visual system (HVS) than previously developed CFAs. Therefore, successfully developing a new image sensor requires the evaluation criteria of the image rendered through the image-processing pipeline. The

**Competing interests:** The authors have declared that no competing interests exist.

pipeline typically consists of demosaicking, noise reduction, white balance, CFA interpolation, color conversion, and gamma correction for rendering the sensor data. However, an overall picture quality assessment of a newly developed CFA often occurs later than the design of the new CFA. In addition, an evaluation of the IQ after the CFA development is conducted by measuring the IQ evaluation items developed thus far, either one by one or in groups. This is because some CFAs are not applicable before the image processing pipeline process, or there are no comprehensive HVS-based IQ evaluation systems after the pipeline. The framework for analyzing the image characteristics of CFAs, the Image Systems Engineering Toolbox (ISET), was first developed by Wandell et al. [2–4]. The ISET is a camera simulation software that receives spectral information from scenes and illuminants, and creates rendered images through optical modeling such as camera lenses and camera sensor simulation. The ISET software has been verified using data of various devices. [5, 6]. In the present study, an IQ analysis of the major CFAs developed to date using existing and newly proposed metrics on the framework was conducted.

Color images are acquired through multiple sensors or a single sensor. Although multiple sensors can acquire high-quality images, they have problems in terms of size and price because they require as many sensors as the number of color planes constituting the color image. As a result, mobile devices acquire color images using a single sensor. To produce color images from a single sensor, an array of color filters is attached over a single sensor [7, 8]. The color arrangement within a single sensor has only a one-color channel signal at a particular location, and the color components of the other two channels are therefore lost. Thus, to obtain a full RGB three-channel color image from a single-color sensor, the two-color channel signals lost at a particular location must be interpolated. This color interpolation process is called demosaicing [7]. In this paper, bilinear [9], laplacian [10], adaptive laplacian [11], projection onto convex sets (POCS) [12] interpolations are considered for evaluating CFAs.

Many CFA patterns with different primary colors have been developed to date, as highlighted in [13], namely, RGB [1, 14–17], RGBE [18, 19], CMY [16, 20–23], and RGBW [24–30] CFAs. The CFA pattern affects the resolution, sharpness, aliasing, reconstruction errors, and dynamic range of the sensor captured image. The captured image is intrinsically affected by an optical feature of the CFA patterns as well as the spectrometry and response characteristic of the image sensor. However, it is also related to the HVS.

The mean squared error (MSE) obtained by averaging the squared intensity differences between an original image and its reproduction is the most widely used image-quality metric (IQM). IQMs based on the MSE are easy to calculate and have obvious physical differences. However, they are not very well matched with the perceived visual quality (e.g., [31–36]). In addition, the root mean square (RMS) metric does not include any information about the device used to present the images. In other words, the RMS error value is not a calibrated value. Because display technologies intrinsically have non-linear transfer functions, the displayed image looks different on a display. For example, each display has a different display gamma. Therefore, un-calibrated images are not suitable for measuring the perceptual difference. In addition, HVS-based IQ evaluation metrics have been studied from the past.

To solve the problem of a non-reflection of the cognitive quality of an MSE-based method, numerous IQMs have been proposed [37, 40–45]. CIELAB for measuring color reproduction errors was designed to approximate human-vision based color discrimination and aspires to achieve perceptual uniformity. However, it turns out that color discrimination is determined by numerous factors, including the spatial pattern of the image and the visual processing [38, 39]. S-CIELAB, a spatial extension of CIELAB, was presented by adding a color separation and spatial filtering procedure to CIELAB to account for human spatial-color sensitivity [40, 41]. Meanwhile, the structural similarity (SSIM) was presented to consider the image degradation

as the perceived change in structural information [42]. The SSIM is based on the assumption that the HVS is highly adapted to extract structural information from the visual field. In addition, the modulation transfer function (MTF) or spatial frequency response (SFR) describes the image resolution and perceived sharpness as the objective assessment of the imaging performance of an optical system [43–45]. Recently, IQ assessment methods closely correlated with HVS, such as MDSI [46] and HaarPSI [47], were presented. MDSI measures the structural and color distortion using combination of gradient similarity (GS) and chromaticity similarity. HaarPSI evaluates local similarities as well as entire similarities using the coefficients obtained from Haar wavelet decomposition.

Gasparini etc proposed a no-reference metric for measuring demosaicing artifacts through psycho-visual experiments [48]. Using a psycho-visual comparison test adopting a single or double stimulus method, it analyzes the subjective evaluation of the demosaicing artifacts. Then, it introduce a no-reference metric for demosaicing artifacts based on measures of blurriness, chromatic and achromatic distortions that are able to fit psycho-visual experiments. While the method focuses on a no-reference metric definition of subjective (perceptual) IQ assessment for demosaicing methods in a given CFA structure, this paper introduces a combination of proven metrics for automatic and objective IQ evaluation for CFA structures as well as demosaicing methods. This paper proposes a CFA IQM system for quantitatively evaluating the existing CFAs or CFAs to be developed in the future. The metrics for measuring the CFA performance include 1) a color error using the CIELAB color metric in the Macbeth Color Checker (MCC), 2) the color reproduction error (visible distortion) using S-CIELAB, 3) the SSIM, 4) the MTF5033 (SLANTED-BAR), 5) the moiré robustness using a MSP [49, 50], 6) an AR error using a GRAY-BAR, 7) structural and color distortion using MDSI, and 8) perceptual similarity using HaarPSI. CIELAB, S-CIELAB, SSIM, MTF50, MDSI, and HaarPSI are existing IQMs, and MSP and AR are newly proposed metrics in this study.

## 2. Materials and methods

### 2.1. Kind of CFAs

Various CFAs used commercially or for research are shown in Fig 1. A mosaic of Bayer CFA arranges RGB color filters on a square grid of photo-sensors, the pattern of which is 50% green, 25% red, and 25% blue [1]. First, the figure includes RGB1 (Yamanaka CFA [14]), RGB2 (Lukac CFA [15]), RGB3 (vertical stripe CFA [16]), RGB4, (diagonal stripe CFA [16]), and RGB5 (modified Bayer CFA [15,17]). There are no known studies addressing the

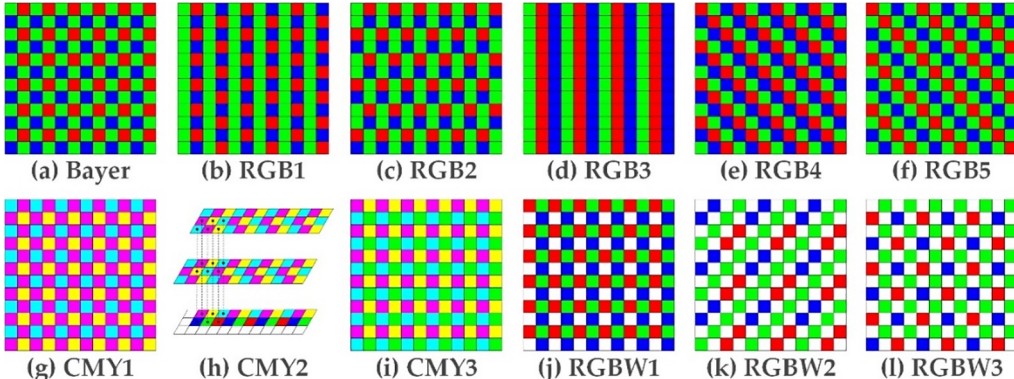

**Fig 1.** Test CFAs for performance comparison: (a)–(f) RGB-, (g)–(i) CMY-, and (j)–(l) RGBW-based CFAs.

performance issues for other RGB CFAs except for Bayer CFA (RGB2–RGB6) in such a comprehensive and systematic manner. In addition, CMY1 uses CFA of secondary colors, again to allow more of the incident light to be detected rather than absorbed [16]. CMY2 (Switchable CMY, RGBCY CFA [20]) has a pair of CMY CFAs that can switch between multiple sets of color primaries (namely, RGB, CMY, and RGBCY) in the same camera. These CFA shift structures and switchable primaries are known to be useful for improving the optimal color fidelity and signal-to-noise ratio in various types of scenes. CMY3 (CGMY CFA [21–23]) is a CFA pattern using subtractive colors, such as cyan, magenta, yellow (C, M, Y), and green to deal with low light conditions. An RGBW1 (RGB and White (W)) matrix is a CFA pattern that includes a white (or transparent or panchromatic) filter element with high sensitivity [24–27]. Panchromatic pixels generate the luminance information, whereas chromatic pixels such as R, G, and B produce the color information. RGBW2 is a CFA in which RGB pixels and panchromatic pixels diagonally alternate in a minimal repeating unit of 4 × 4 pixels [28, 29]. RGBW3 [30] has first and second lines, which filter elements for luminance components disposed in each line and are offset from the filter elements for the luminance components in an adjacent line, where the first line includes filter elements for two-color components, and the second line includes filter elements for a single-color component.

## 2.2. Proposed CFA IQ evaluation system

To simulate the proposed CFA IQ evaluation system, we used ISET [5,6] with bilinear [9], laplacian [10], adaptive laplacian [11], and POCS [12] demosaicing. As mentioned in section 1, the proposed CFA evaluation system consists of eight metrics for respective evaluations of the color accuracy, color reproduction, structural information, image contrast, moiré phenomenon, and noise. Fig 2 shows the imaging pipeline for the proposed CFA IQ evaluation system. In the proposed system, the CFA structure and demosaicing method are changeable, and the CFA IQ evaluation results are plotted on the polar coordinates.

Fig 3 shows the test input images used for the proposed CFA IQ evaluation system: a) SLANTED-BAR (ISO 12233 resolution chart) [51] for calculating the image contrast using MTF50, b) MCC [52] for measuring the color error using CIELAB, c) PUPPY [53] for measuring the color reproduction error (visible distortion) using S-CIELAB, analyzing the structural information using SSIM, structural and color distortion using MDSI, and perceptual similarity using HaarPSI, d) LINEAR-CHIRP for analyzing the moiré robustness using MSP, and e) GRAY-BAR for analyzing the noise robustness. Applying an appropriate and high-quality initial dataset is essential to accurately assessing the system performance. Of the test input images, PUPPY is the only multipectral scene. The sensor response of multispectral scenes is calculated, then CIE XYZ value at each pixel location is computed by the ISET camera simulator.

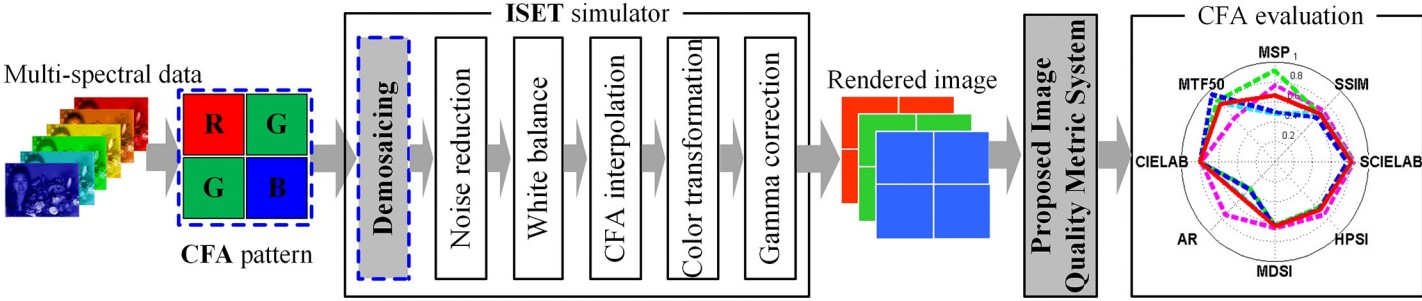

**Fig 2. Proposed CFA IQ evaluation system.**

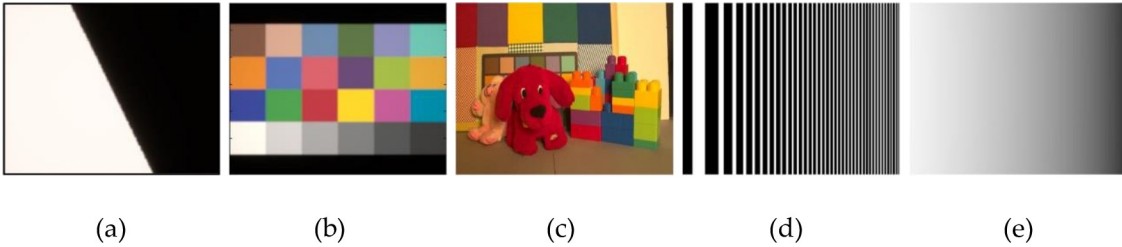

**Fig 3.** Test input images used in the proposed CFA IQ evaluation system: (a) SLANTED-BAR, (b) MCC, (c) PUPPY, (d) LINEAR-CHIRP, and (e) GRAY-BAR.

MCC color image was created based on the Gretag-MCC [52]. The rest of the test input images are color images created by patterns generated by the algorithm. PUPPY has a 32-bit resolution and a spectral wavelength of 400–700 nm, and are illuminated by a D65 illuminant with a mean luminance of 100 cd/m$^2$ (varying mean luminance with 3, 6, 12, 50, 100, 200, 400 cd/m$^2$ for the S-CIELAB and SSIM experiments using PUPPY). The field of view (FOV) is 2˚ for a SLANTED-BAR, 30˚ for MCC, and 10˚ for PUPPY, LINEAR-CHIRP, and GRAY-BAR. The resolution is 636 × 720 for SLATED-BAR, MCC, and PUPPY, and 500 × 500 for LINEAR--CHIRP and GRAY-BAR.

**2.2.1. Color error using CIELAB with MCC.** The Delta E metric, calculated by CIE $L^*a^*b^*$, is one of the extremely well-known perceptual color fidelity metrics [37]. The spectral power distribution derived from the radiant power emitted by two light sources is transformed into CIE XYZ values. The CIE XYZ represents the spectral sensitivity of the three types of cone cells that are sensitive to the RGB primary colors. This means that the CIE XYZ values are a device-invariant representation of color. The CIE XYZ values are transformed into a $L^*a^*b^*$ space, in which an equal perceptual color difference corresponds to an equal distance. The perceptual color difference between the reference (ideal) image and the rendered image can then be calculated by taking the Euclidean distance of the two images in the $L^*a^*b^*$ space [54]. The color difference is represented by $\Delta E^*$ units. To evaluate the color accuracy of an ideal image (using an ideal sensor) and a CFA output image, MCC shown in Fig 3(B) is rendered under the D65 illuminant used in the proposed system. In addition, CIELAB $\Delta E^*$ is calculated as the Euclidean distance between two colors. Three numerical values are $L^*$ for the lightness and $a^*$ and $b^*$ for the green–red and blue–yellow color components. The metrics for the MCC patches include the color error, the lightness error for the six gray patches, and the $xy$ chromaticity.

**2.2.2. Color reproduction error (visible distortion) using S-CIELAB.** S-CIELAB Delta E describes how a spatial pattern causes a visual difference based on the assumption of a color-pattern separability, whereas the CIELAB Delta E metric estimates the magnitude of the difference between two color stimuli in a uniform color space. To apply the CIELAB Delta E metric to color images, the spatial patterns of the image are considered [40, 41, 55]. Fig 4 shows the S-CIELAB procedure. The S-CIELAB includes the color separation and spatial-filtering process convolving with kernels of different sizes and shapes before the CIELAB step. The S-CIE-LAB Delta E metric extends the CIELAB to include the spatial sensitivity, and represents the visibility of the distortion in an image.

S-CIELAB is largely composed of three steps. The first step is a color separation step, in which the original (ideal) image and test (rendered using a CFA) image are transformed into the luminance, red/green, and blue/yellow components [56]. The second step is a spatial filtering step in which the respective separated components are filtered using spatial filters based on the spatial sensitivity of the human eye. Finally, the third step is the CIE-XYZ transformation

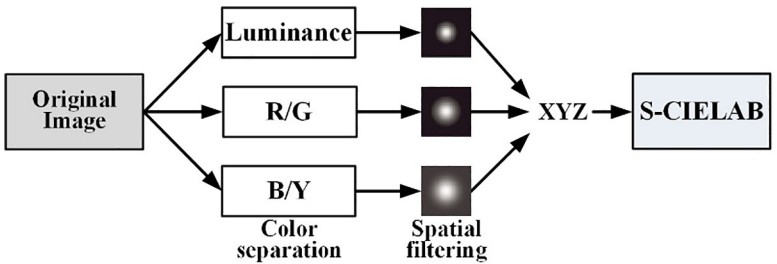

**Fig 4. S-CIELAB model.**

of the filtered components before the CIELAB step. S-CIELAB can obtain a $\Delta E_s^*$ map using the CIELAB color difference equation. The error map describes where the test image is visually distorted as compared to the original image.

The S-CIELAB difference between the original (ideal) image and the test (rendered) image estimates the reproduction error and visual distortion. Except for the three steps added in S-CIELAB, the reproduction error calculation of S-CIELAB is the same as that of CIELAB. The S-CIELAB difference describes the spatial sensitivity as well as the color sensitivity. The S-CIELAB difference is the same as the CIELAB for a uniform region, although it finds a visual difference more accurately than the CIELAB for a complex pattern region. The color difference of S-CIELAB is expressed in $\Delta E_s^*$ units in this study. To evaluate the color reproduction of a CFA output image and an ideal reproduction, we use the PUPPY image shown in Fig 3 (C).

**2.2.3. Structural information using SSIM.** Under the hypothesis in which human visual perception (HVS) is greatly adapted to retrieve structural information from a scene, SSIM was devised for a quality assessment based on a degradation of the structural information [42]. Specifically, the SSIM index is used for measuring the similarity between the original (ideal) and target images (output images by CFA). The peak signal-to-noise ratio and MSE based metrics do not reflect HVS. By contrast, the SSM takes account the perceived image degradation based on the loss of structural information. The structural information of the image signifies the strong dependencies between pixels owing to their spatial closeness. Such spatial dependencies maintain significant information regarding the structure of an object in any visual scene.

The system diagram of the SSIM is shown in Fig 5. The SSIM system separates the similarity measurement into three comparisons: luminance, contrast, and structure. For $x$ (ideal image) and $y$ (CFA output image), two nonnegative image signals, a luminance comparison function $l(x,y)$, a contrast comparison function $c(x,y)$, and a structural comparison function $s(x,y)$ are

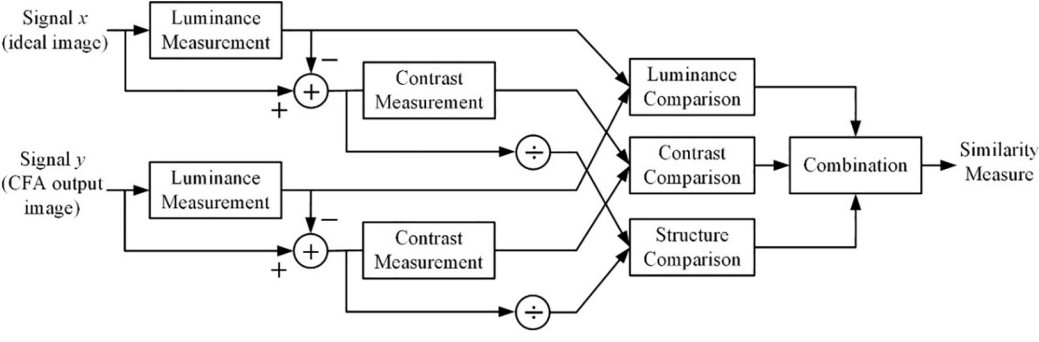

**Fig 5. SSIM procedure.**

calculated. Then, the three comparisons are combined, and the SSIM index between image $x$ and $y$ is obtained as follows:

$$SSIM(x, y) = [l(x, y)]^{\alpha} \cdot [c(x, y)]^{\beta} \cdot [s(x, y)]^{\gamma}. \tag{1}$$

In the case of $\alpha = \beta = \gamma = 1$, a specific form of the SSIM index is as follows:

$$SSIM(x, y) = \frac{(2\mu_x\mu_y + c_1)(2\sigma_{xy} + c_2)}{(\mu_x^2 + \mu_y^2 + c_1)(\sigma_x^2 + \sigma_y^2 + c_2)}. \tag{2}$$

where constants $C_1 = (K_1 L)^2$ and $C_2 = (K_2 L)^2$. In addition, $I$ is the dynamic range of the pixel values (255 for 8-bit grayscale images) and $K_1 \ll 1$, $K_2 \ll 1$. To evaluate the structural information of an ideal image and the output images by the CFAs, the PUPPY image shown in Fig 3(C) is used.

**2.2.4. Image contrast using MTF50 by slanted-bar.** The MTF is calculated using ISO 12233 (slanted-bar) with ISET [41, 42]. The MTF accurately describes the image contrast attenuation for each spatial frequency. To obtain the MTF of an imaging system, the ISO 12233 examines a slanted edge for all color channels [45]. The luminance MTF can be derived by combining the respective MTF for all color channels. The edge response function can be expressed as the integrated line spread function (LSF) through a differentiation. A Fourier transformation of the LSF provides the corresponding MTF. Accordingly, the method analyzes the edge response for computing the MTF through the LSF.

The slanted-bar method specified in ISO 12233 integrates the line measurements at the edge location. The measurements solve the down-sampled trouble of the imaging system using a super-sampled edge. Fig 6(A) shows a rectangular region near the slanted edge. The derivative for horizontal lines for all color channels is integrated into the edge response of the imaging system. Through the edge response, the luminance MTF of the system as well as the LSF and MTF for all color channels are derived.

Fig 6(B) shows the MTF for all color channels and the luminance MTF of the system. The horizontal and vertical axes indicate the spatial frequency in cycles per millimeter at the sensor surface and the contrast reduction, respectively. The contrast reduction on the vertical axis represents the SFR in the ISO standard. The red, green, blue, and black lines represent the MTF for the R, G, and B channel and the luminance MTF, respectively. The luminance MTF is calculated by the weighted sum (luminance = 0.3R + 0.6G + 0.1B) of the respective color channels. In addition, the Nyquist sampling frequency (cycles/mm), MTF50, and percent alias are shown in the figure. The MTF50 indicates the spatial frequency where the luminance MTF

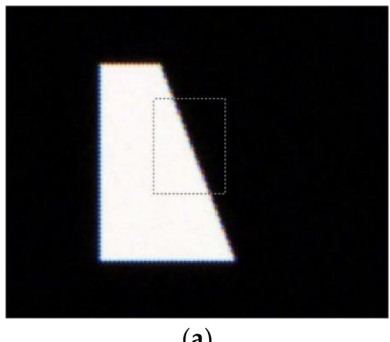

(a)

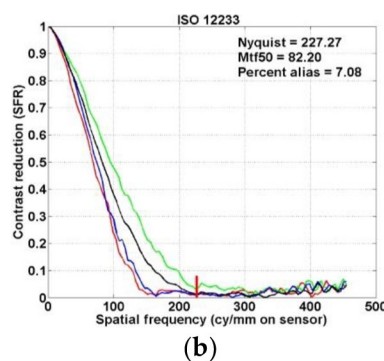

(b)

**Fig 6.** (a) Created slanted-bar and (b) its corresponding MTF50 curve.

becomes 0.5. In addition, the percent alias, namely, the percentage of aliasing, is calculated as the area under the right luminance MTF at the Nyquist frequency. The Nyquist frequency is indicated by the vertical red line in the figure. To analyze the image contrast for a rendered image using a CFA, the SLANTED-BAR image shown in Fig 3(A) is applied.

**2.2.5. Moiré robustness using MSP through a linear chirp.** CFA-output images can be degraded by the appearance of a moiré pattern occurring in the digital imaging system. A color moiré has artificial color banding that can appear in images with repetitive patterns of high spatial frequencies [49, 50]. The color moiré is the result of aliasing (image energy above the Nyquist frequency) in an image sensor. It is actually difficult to quantitatively estimate a moiré phenomenon because it is spatially irregular and its color band is varied. Thus, we use a linear-chirp pattern (gradually narrowing the widths of the black and white stripes) with a low to high spatial frequency for quantitatively estimating the moiré robustness. The linear-chirp signal is a sinusoidal wave that increases linearly in terms of frequency. Because a moiré phenomenon is an unintended color band, we analyze it using the square root of the sum of the square of only $a^*$ and $b^*$, $\sqrt{(a^*)^2 + (b^*)^2}$ (an $ab$ color value is considered a moiré) for the central horizontal line in the linear-chirp pattern. In addition, it is filtered using a one-dimensional ($1 \times 5$) mean filter to reduce the surrounding noises. Then, a moiré value of higher than a threshold is regarded as the MSP toward a low frequency to a high frequency, because tiny moiré in a low-frequency region does not affect the human eye.

Fig 7 shows an example of a moiré measurement using a linear-chirp pattern. We can see that an unintended color band occurs in a high-frequency region of the CFA output image in Fig 7(A), even though the input linear-chirp pattern shown in Fig 3(D) consists only of black and white stripes. Fig 7(B) shows an $ab$ color image for the CFA output image. In the $ab$ color image, the low-frequency region has little unintentional color value and thus has little moiré phenomenon, whereas the high color value in the high-frequency region means that the moiré

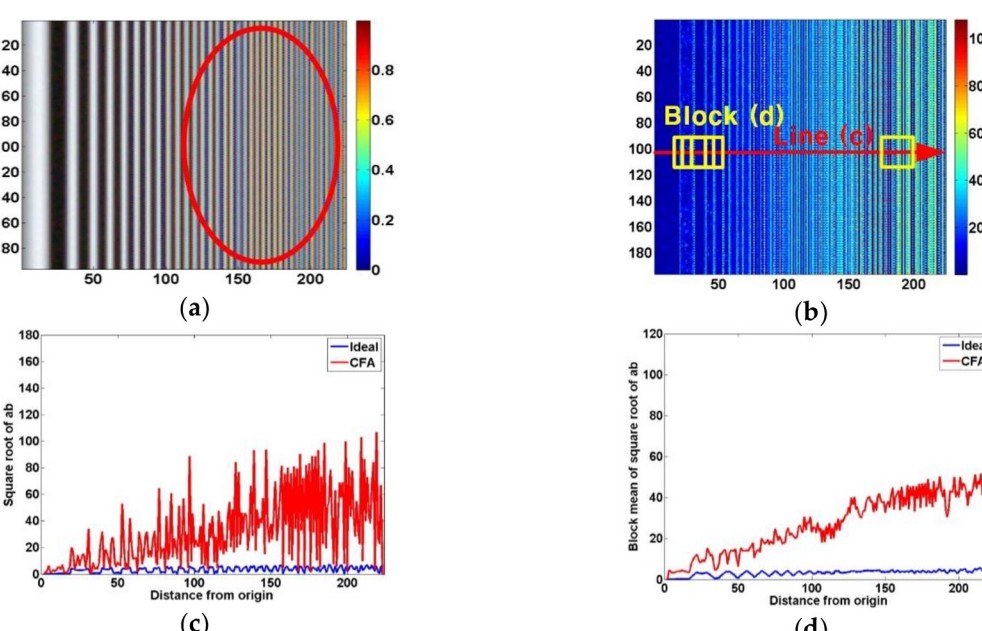

**Fig 7.** (a) CFA output image for LINEAR-CHIRP image, (b) $ab$ color ($\sqrt{(a^*)^2 + (b^*)^2}$) image for CFA output image, (c) color value for central horizontal line, and (d) mean color value of $5 \times 5$ block for central horizontal direction on $ab$ color image.

phenomenon is severe. Fig 7(C) and 7(D) represent the color value for the central horizontal line and the average color value of a 5 × 5 block for the central horizontal direction on the *ab* color image. We can confirm that the moiré value becomes higher toward a high frequency. As shown in Fig 7(C), the moiré is extremely irregular in nature regardless of the frequency band. Nevertheless, it can be seen in Fig 7(D) that the moiré increases gradually from low frequency to high frequency through the average color value of the block for the central horizontal direction. The blue and red curves indicate the respective resulting moiré curves by the ideal sensor and arbitrary CFA. We increased the block size to 15 × 15 to better understand the moiré characteristics of each CFA. Using a threshold for the average color value of the block, the MSP for each CFA is analyzed. In addition, the MSP is expressed in units of spatial frequency (cycles per degree).

**2.2.6. Achromatic reproduction using gray-bar.** The AR error of achromatic color is measured by the difference in luminance for the central horizontal line of the ideal and CFA output images for GRAY-BAR shown in Fig 3(E). The brightness of the GRAY-BAR decreases for the vertical line in the image. The original GRAY-BAR image is degraded owing to the change in luminance and noise during the rendering process through the CFA. The luminance value of the central horizontal line of the output image rendered by an arbitrary CFA for the GRAY-BAR contains a partial luminance change and white noise compared with the output image by the ideal sensor. The AR error for any CFA is measured by looking at the average difference in luminance for the central horizontal line between the GRAY-BAR rendered by an ideal sensor and the CFAs.

**2.2.7. Structural and color distortion using MDSI.** The MDSI utilizes gradient and chrominance features to measure structural and color distortion [46]. A gradient-chromaticity similarity map is made by combining these two similarity maps. First, for reference (ideal) and distorted (CFA) images, $R$ and $D$, GS is obtained by

$$\text{GS}(\text{x}) = \frac{2G_R(x)G_D(x) + C_1}{G_R^2(x) + G_D^2(x) + C_1},\tag{3}$$

where $C_1$ is a constant to control numerical stability. The gradient-color similarity (GCS) is calculated as the follows:

$$\overline{\text{GCS}}(\text{x}) = \alpha\overline{\text{GS}}(\text{x}) + (1 - \alpha)\overline{\text{CS}}(\text{x}),\tag{4}$$

where $\overline{\text{GS}}(\text{x})$ and $\overline{\text{CS}}(\text{x})$ means the enhanced gradient and color similarity function [46]. And the MDSI is defined as the follows:

$$\text{MDSI} = \left[\frac{1}{4}\sum_{i=1}^{N}|\overline{\text{GCS}}_i^{1/4} - \left(\frac{1}{N}\sum_{i=1}^{N}\overline{\text{GCS}}_i^{1/4}\right)|\right]^{1/4},\tag{5}$$

**2.2.8. Perceptual similarity using HaarPSI.** The HaarPSI was presented for yielding full reference IQ assessments [47]. The HaarPSI evaluates local similarities as well as entire similarities between two images by using the coefficients obtained from a Haar wavelet decomposition. For two grayscale images $f_1$, $f_2$, the local similarity is computed based on a 2D discrete Haar wavelet transform as the following;

$$\text{HS}_{f_1,f_2}^{(k)}[x] = l_\alpha\left(\frac{1}{2}\sum_{j=1}^{2}\text{S}(|(g_j^{(k)} * f_1)[x]|, |(g_j^{(k)} * f_2)[x]|, C)\right),\tag{6}$$

where C>0, k∈{1,2} selects either horizontal or vertical Haar wavelet filters and S represents

the similarity. The HaarPSI is computed by

$$\text{HaarPSI}_{f_1,f_2} = l_\alpha^{-1} \left( \frac{\sum_x \sum_{k=1}^{2} \text{HS}_{f_1,f_2}^{(k)}[x] \cdot \text{W}_{f_1,f_2}^{(k)}[x]}{\sum_x \sum_{k=1}^{2} W_{f_1,f_2}^{(k)}[x]} \right)^2 , \qquad (7)$$

where W means a weight map which is derived from the response of a single low-frequency Haar wavelet filter.

**2.2.9. Normalization of metrics and its combination.** The polar coordinate was selected to observe the entire performance of the test CFAs. To quantitatively and visually evaluate the test CFAs, the measured eight metrics should be normalized with the dynamic range of [0, 1]. The smaller the value of the color difference $\Delta E$ by CIELAB with the MCC image is, the color difference (color reproduction error) $\Delta E_s$ by S-CIELAB and MDSI with the PUPPY image, and the AR error $\Delta N$ with the GRAY-BAR image, the better the reproduction performance and structural (and color) similarity for the color and luminance by the CFAs. By contrast, the larger the SSIM and HaarPSI value from the PUPPY image, the MTF50 value from the SLAN-TED-BAR image, and the MSP value from the LINEAR-CHIRP image are, the better the structural information preservation, perceptual similarity, image contrast, and moiré robustness performance of the CFAs. First, $\Delta E$, $\Delta E_s$, and $\Delta N$ measured using CIELAB, S-CIELAB, AR, and MDSI are normalized to give a higher score to the smaller difference value as in the following:

$$\begin{aligned}
n\Delta E &= (\Delta E_{\max} - \Delta E)/(\Delta E_{\max} - \Delta E_{\min}) \\
n\Delta E_s &= (\Delta E_{s.\max} - \Delta E_s)/(\Delta E_{s.\max} - \Delta E_{s.\min}) \\
n\Delta N &= (\Delta N_{\max} - \Delta N)/(\Delta N_{\max} - \Delta N_{\min}) \\
n\text{MDSI} &= (\text{MDSI}_{\max} - \text{MDSI})/(\text{MDSI}_{\max} - \text{MDSI}_{\min})
\end{aligned} \qquad (8)$$

where the min and max values of each of the above metrics are $\Delta E_{\max} = 6$, $\Delta E_{\min} = 2.2$, $\Delta E_{s.\max} = 9.5$, $\Delta E_{s.\min} = 2.5$, $\Delta N_{\max} = 1.0$, $\Delta N_{\min} = 0.5$ and $\text{MDSI}_{\min} = 0.5$, $\text{MDSI}_{\min} = 0$. By contrast, the measured SSIM, MTF50, MSP, and HaarPSI for the structural information, image contrast, and moiré starting point are normalized to give a higher score to a bigger measurement value, as in the following:

$$\begin{aligned}
n\text{SSIM} &= (\text{SSIM} - \text{SSIM}_{\min})/(\text{SSIM}_{\max} - \text{SSIM}_{\min}) \\
n\text{MTF50} &= (\text{MTF50} - \text{MTF50}_{\min})/(\text{MTF}_{\max} - \text{MTF}_{\min}) \\
n\text{MSP} &= (\text{MSP} - \text{MSP}_{\min})/(\text{MSP}_{\max} - \text{MSP}_{\min}) \\
n\text{HaarPSI} &= (\text{HaarPSI} - \text{HaarPSI}_{\min})/(\text{HaarPSI}_{\max} - \text{HaarPSI}_{\min})
\end{aligned} \qquad (9)$$

where the min and max values of the respective metrics are $\text{SSIM}_{\max} = 1$, $\text{SSIM}_{\min} = 0$, $\text{MTF50}_{\min} = 110$, $\text{MTF}_{\min} = 30$, $\text{MSP}_{\max} = 150$, $\text{MSP}_{\min} = 20$, and $\text{HaarPSI}_{\max} = 1$, and $\text{HaarPSI}_{\min} = 0$. The SSIM is originally calculated as [0, 1]. The min and max values used to calculate the score of each metric were empirically derived by considering the distribution of the measured values of the test CFAs for each metric.

## 3. Results and discussion

We simulated the performances of all test CFAs with eight metrics with the text input images. The bilinear, laplacian, adaptive laplacian, and POCS demosaicing method for each CFA is used in the proposed system. Figs 8 to 15 and Table 1 show comparisons between test CFAs by

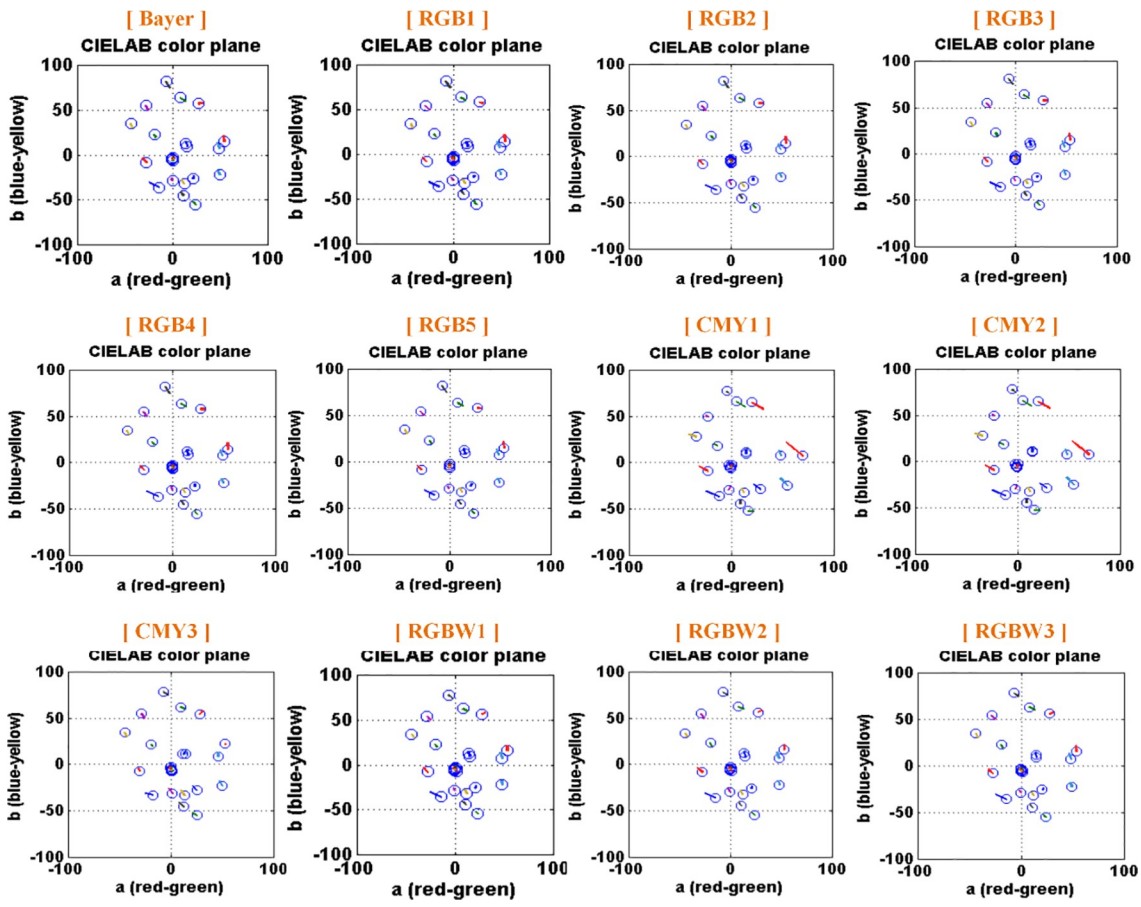

**Fig 8. Color difference error of 24 color patches for test CFAs.**

bilinear demosaicing, while Figs 16 and 17 and Tables 2 and 3 show overall comparisons between test CFAs by all the demosacings used in this paper.

Fig 8 shows the color difference for the color patches for the test CFAs using the CIELAB metric and MCC. In the CIELAB plane, the blue circles indicate the measured value, and the small red lines show the distance from the measurement to the ideal value. All of the test CFAs commonly show larger color difference in the blue color region. In addition, CMY1 and CMY2 show large color difference even in the red-yellow region. In the mean Delta E of Table 1, the RGBW CFAs showed the smallest color difference of (2.58~2.63), and the color difference of the RGB CFAs (2.93~2.97) were smaller than the CMY CFAs (2.61~3.70). It can be seen that the RGB CFAs have larger color difference compared with the RGBW or CMY CFAs. In the mean Delta L for the lightness (luminance) error, the RGBW CFAs were the smallest at 2.13–2.16, and the RGB CFAs have the highest lightness error at 4.01~−4.06, whereas CMY CFAs range at 3.86~−3.97.

Fig 9 shows the chromaticity of test CFAs for the MCC. As in the CIELAB color plane of the test CFAs mentioned above, all test CFAs have larger color difference in greenish-blue region even in the chromaticity. Bayer and RGB CFAs have larger color difference in red (or pink) and greenish-blue region, whereas CMY1 and CMY2 among the CMY CFAs have larger color difference in yellow, red, and greenish-blue region. We can see that the CMY3- and RGBW-CFAs among the test CFAs have a relatively smaller color difference for all colors of the MCC.

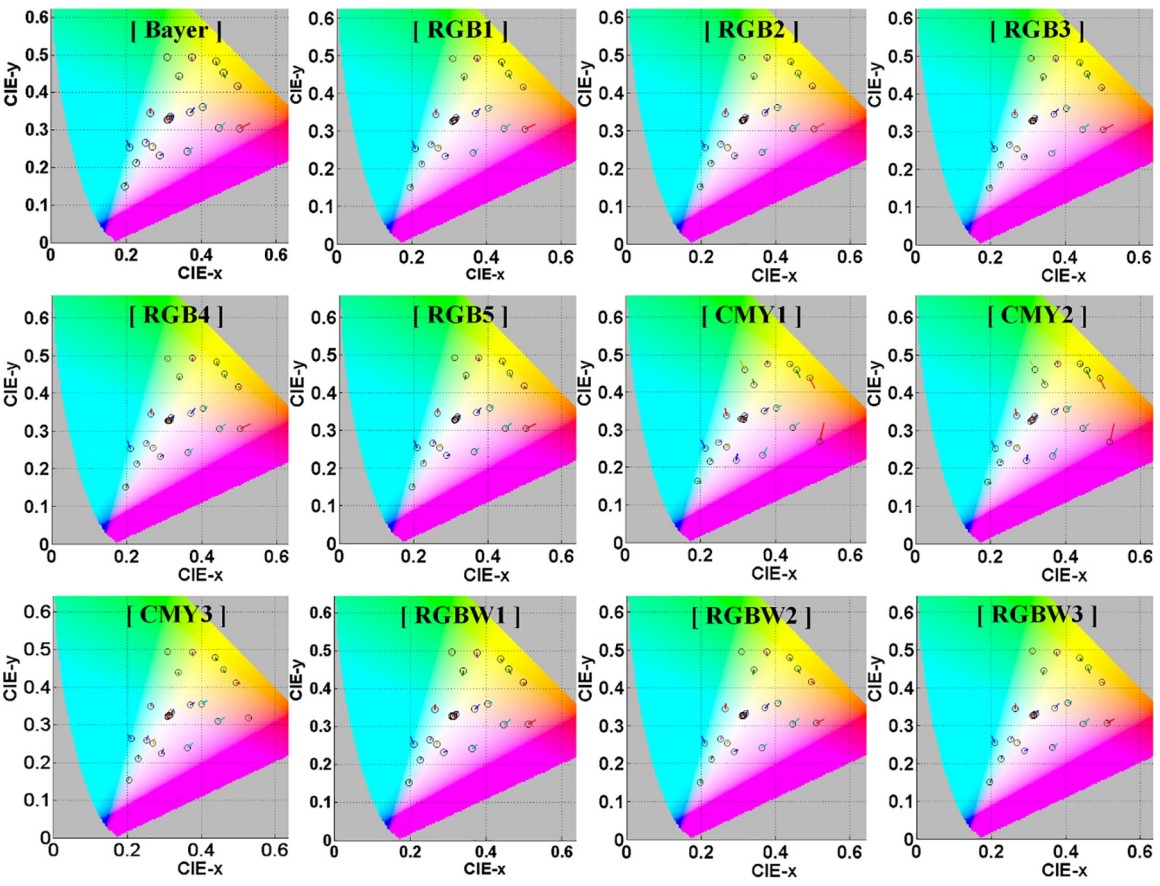

**Fig 9. *xy* chromaticity diagram of test CFAs for MCC.**

Fig 10 shows the color reproduction error ($\Delta E_s^*$) using S-CIELAB and the structural information results using the SSIM for the test CFAs. S-CIELAB typically predicts a lower visibility of the color differences for textured regions. Qualitatively, these predictions are consistent with the measurements of human spatial-color sensitivities. In the color reproduction error, RGB CFAs (3.24~3.50) performed better than the CMY CFAs (5.28~6.09) and RGBW CFAs (3.61~3.80). All of the CFAs commonly have a remarkable color difference for blue (see the blue block region of the lower-right corner in the respective output images). In addition, CMY1 and CMY2 show a particularly larger color difference even in the red (puppy doll region in the output images) and yellow (yellow panel regions in the upper-left corner in the output images) color region. These results are similar to the color error result of CIELAB described above because S-CIELAB is consistent with the basic CIELAB calculation for large uniform areas.

Because the SSIM measures the structural similarity of the luminance, contrast, and structure between an ideal image and the output image, the closer it is to 1, the better the IQ of the output image is, which is contrary to CIELAB and S-CIELAB. All test CFAs obtained almost equally excellent SSIM values (0.73~0.79) except for CMY3. In the results of CIELAB and S-CIELAB, CMY3 showed better (smaller) color error and color reproduction error performance for bluish-green or greenish-blue (upper-center panel region in the output images) color. So we can deduce that CMY3 obtained a poor SSIM value because the structural comparison function among the similarity functions of the SSIM has a lower value (i.e., poor

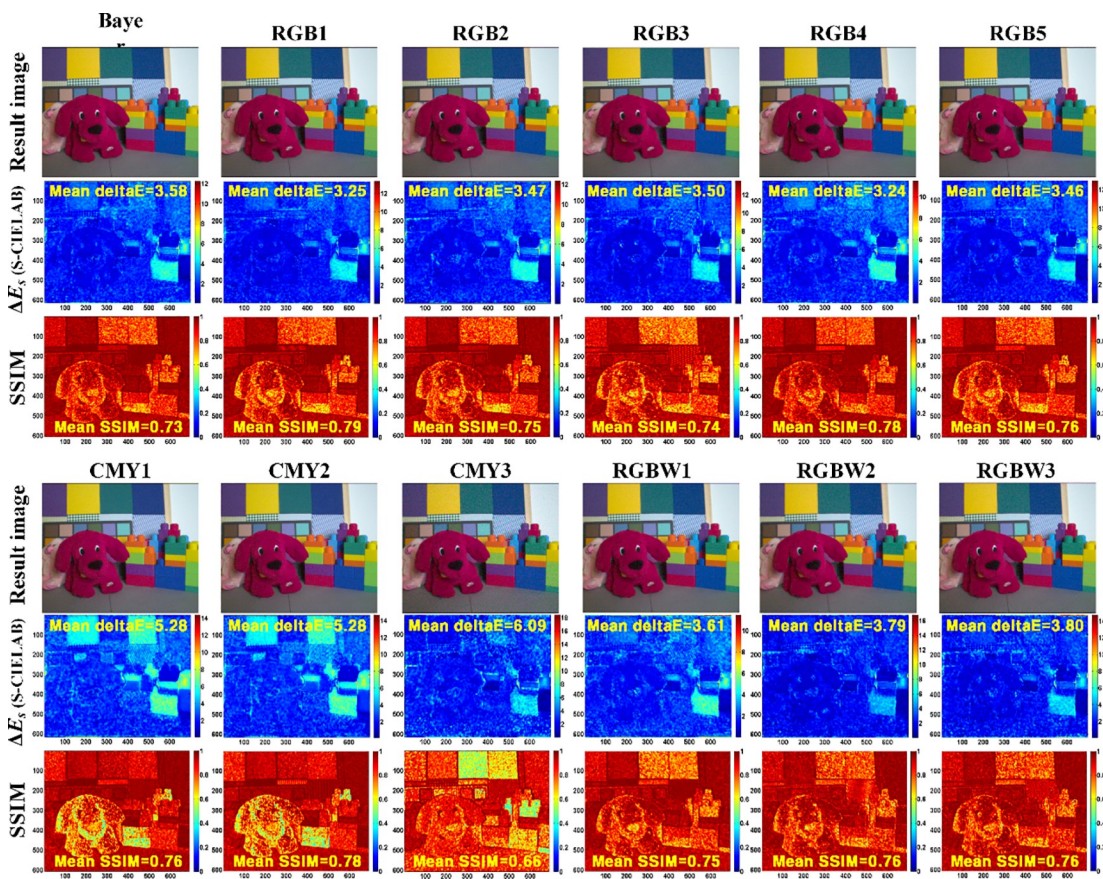

**Fig 10. Color reproduction error using S-CIELAB and structural information results using SSIM for test CFAs.**

structural similarity), compared to the luminance comparison function or the contrast comparison function.

Fig 11 shows the MTF50 of R, G, B, and K (black) color for the test CFAs. The MTF measures the contrast reproductivity, which is the ability to resolve the black and white vertical lines in the rendered SLANTED-BAR image. In general, the contrast reproducibility decreases as the spatial frequency increases. In the figure, the small vertical red line (at 227 cycles/mm) indicates the Nyquist frequency.

As described in section 2.2.4, the luminance MTF indicates that the G color among the RGB colors has the largest weight. As a result, the overall distribution of the luminance MTF for K color is similar to the MTF distribution of the G color. On the other hand, the weights of the R and B color are smaller than that of the G color. Additionally, the MTF distributions of these two colors are similar to each other. Although a CFA is composed of the same elements, the MTF results vary depending on the location and structure of the elements. In the MTF result for R and B color, RGB1, RGB2, and RGB4 among the RGB CFAs, CMY1 and CMY2 among the CMY CFAs, and RGBW1 among the RGBW CFAs show a significant contrast reproductivity. In the MTF results for the G color, RGB1, RGB2, RGB4, and RGB5 among the RGB CFAs, CMY2 and CMY3 among the CMY CFAs, and RGBW2 and RGBW3 among the RGBW CFAs show a superior resolution capability. As a result, we confirmed that Bayer (79.20), RGB1 (75.00), RGB2 (77.60), RGB5 (75.20), CMY2 (78.80), and RGBW1 (74.20) show an excellent contrast reproducibility according to the spatial frequency.

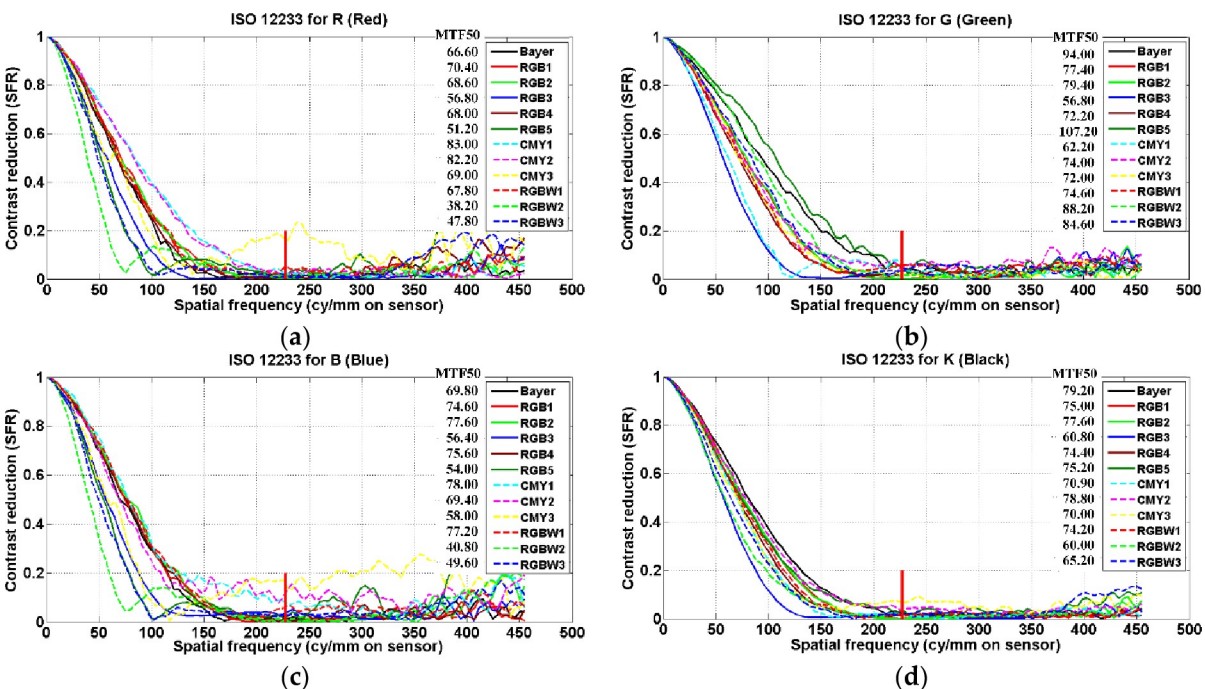

**Fig 11.** MTF50 of (a) R (red), (b) G (green), (c) B (blue), and (d) K (black) color for test CFAs.

During the simulation of the moiré phenomenon, the illuminant used was D65, and the mean luminance was set to 100 cd/m². Fig 12 shows the image output by the test CFAs, where $ab$ ($\sqrt{(a*)^2 + (b*)^2}$) calculates the color of the output image, the original color values for the center-horizontal line of the $ab$ color image, and the filtered color values using a one-dimensional ($1 \times 5$) mean filter for the original color values of the center horizontal line. The test CFAs cause unintended color bands or moiré phenomena within the high spatial frequency region of the sinusoidal, or a linear-chirp pattern. The LINEAR-CHIRP image does not originally contain any color. However, rendering using the CFA causes unintended color to appear in the high spatial frequency band. In the $ab$ color image, the darker red region indicates that the moiré is severe. The original color value for the central horizontal line of the $ab$ color image contains a lot of noise owing to the high-frequency effect. To analyze the moiré color band more quantitatively, a one-dimensional $1 \times 5$ mean filter was applied to the original color value for the central horizontal line of the $ab$ color image. Bayer, RGB2, RGB4, RGB5, and RGBW1 show gentle color values rising from the low to high spatial frequency band, which indicates robustness to the moiré, as compared to the other test CFAs. It is also noteworthy that, even though all CFAs include the same elements, the moiré pattern differs depending on the location and structure of the elements. RGB1 and RGB3 in the RGB CFAs, CMY2 and CMY3 among the CMY CFAs, and RGBW2 and RGBW3 among the RGBW CFAs have already caused a moiré phenomenon even within a relatively lower spatial frequency band than the other test CFAs.

Fig 13 shows the mean color value curves of the central horizontal line of the $ab$ color image using (a) $1 \times 5$ and (b) $1 \times 15$ mean filters, and the MSP results for the test CFAs. As shown in the Fig 13(A), as the spatial frequency increases, the moiré worsens. To more quantitatively evaluate the moiré characteristics, we plotted the mean color value curve using $1 \times 15$ mean filter as shown in Fig 13(B). In the figure, the small red horizontal line indicates the

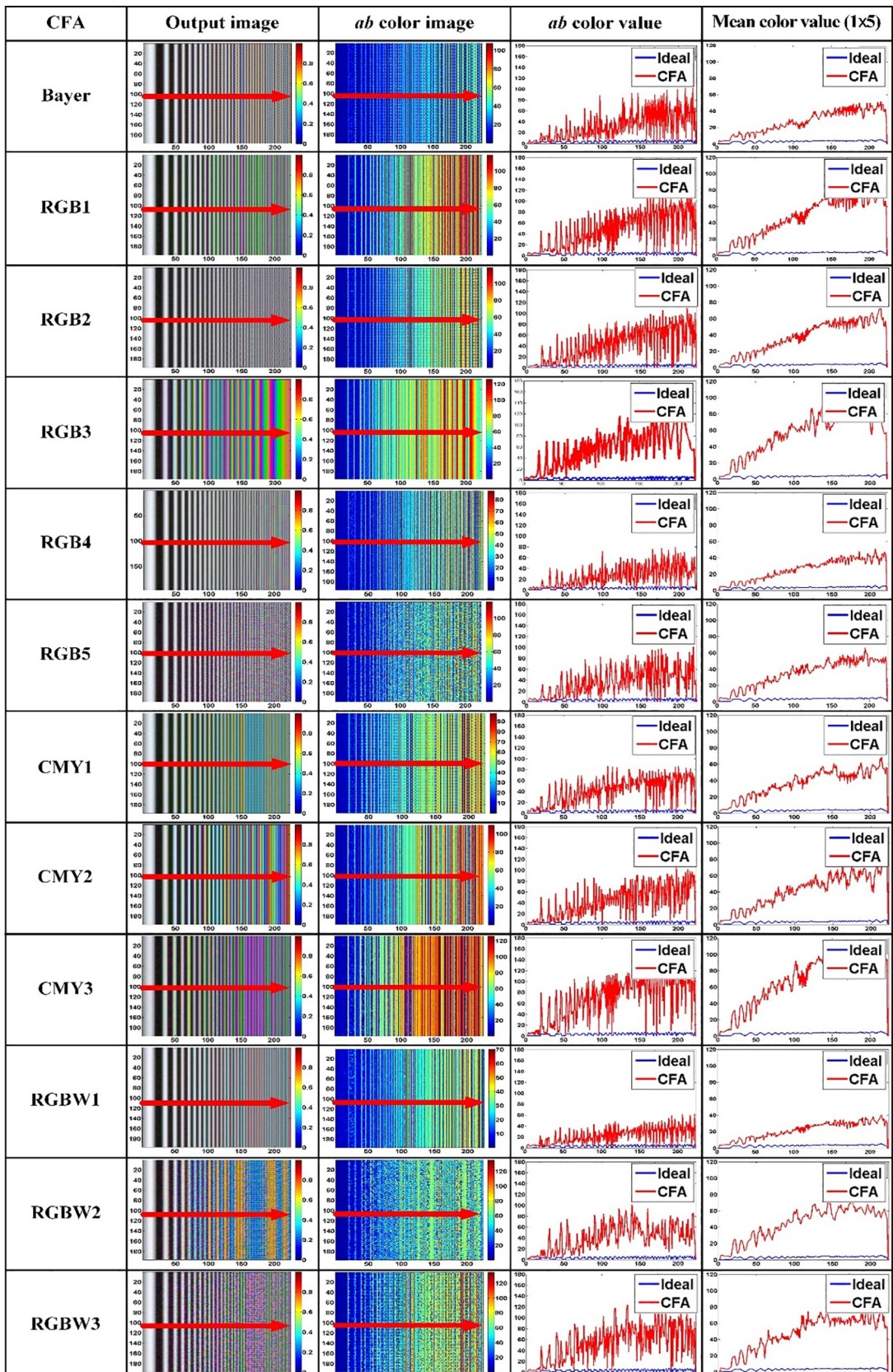

**Fig 12. Output image of test CFAs for LINEAR-CHIRP image,** *ab* **color image for output image, color value for center horizontal line of color image, and filtered color value using one-dimensional (1 × 5) mean filter for the color value.**

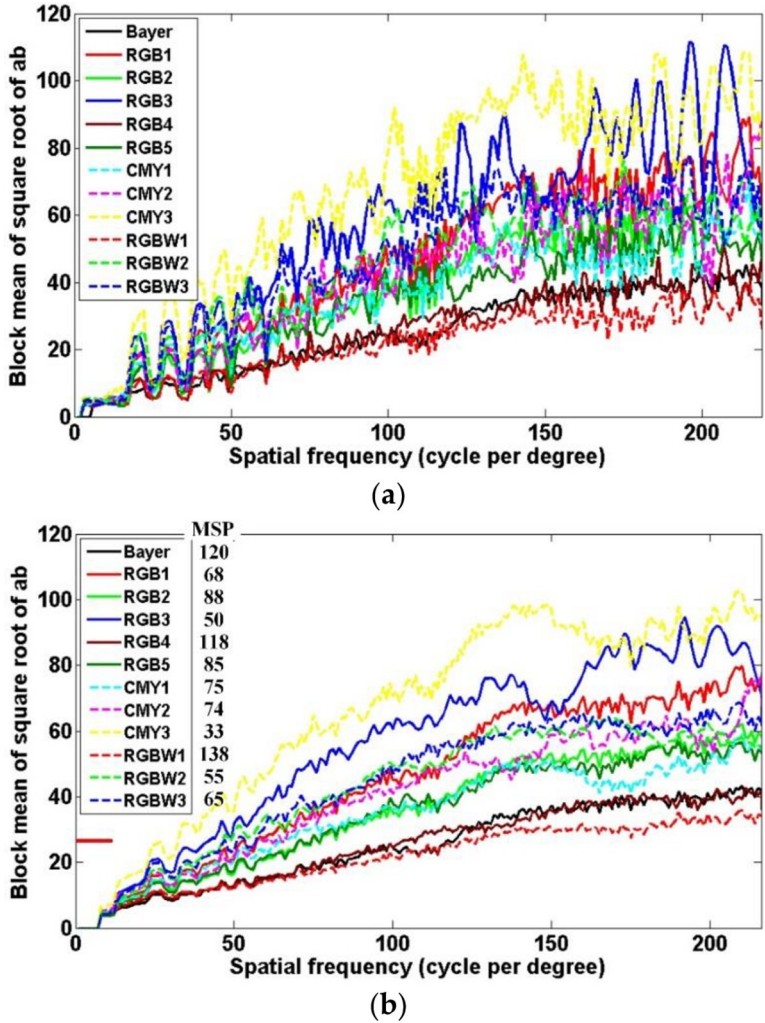

**Fig 13.** Mean color value curves using (a) 1 × 5 and (b) 1 × 15 mean filters and MSP results for test CFAs.

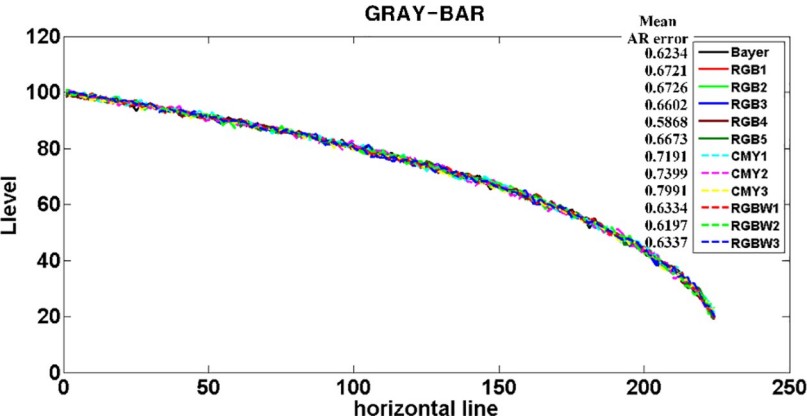

**Fig 14. Luminance value curve and mean AR error value for test CFAs.**

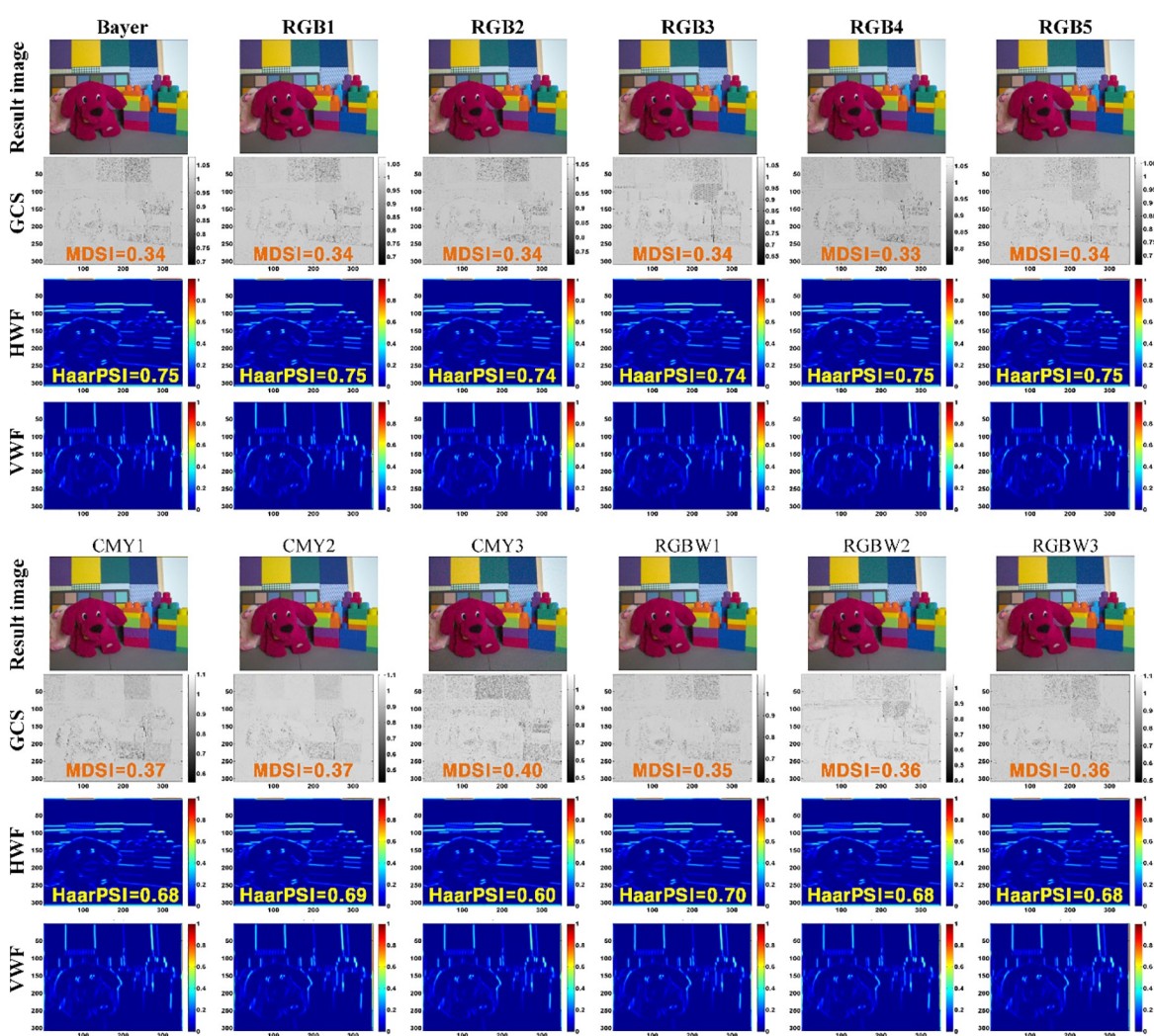

**Fig 15. MDSI and HaarPSI results for test CFAs.**

threshold applied to the filtered color value for detecting the MSP where the moiré starts to occur. The threshold was empirically set at 25. While Bayer (120 cpd), RGB4 (118 cpd), and RGBW1 (138 cpd) show higher MSP performance, RGB3 (50 cpd), CMY3 (33 cpd), and RGBW2 (55 cpd) show a very poor MSP performance.

Fig 14 shows the luminance value curve for the center horizontal line of the output image. The respective output gray-bar images rendered by each test CFA contains some noise. However, the overall condition is excellent. We can see the mean AR error value (i.e., mean noise value), namely, the mean of the absolute difference of the luminance values of the central horizontal line for the gray-bar images rendered by the ideal imaging system and the test CFAs. It

**Table 1. Mean delta E for 24 color patches and mean delta L for 6 achromatic patches for test CFAs.**

| Metrics | Bayer | RGB1 | RGB2 | RGB3 | RGB4 | RGB5 | CMY1 | CMY2 | CMY3 | RGBW1 | RGBW2 | RGBW3 |
|---------|-------|------|------|------|------|------|------|------|------|-------|-------|-------|
| Mean $\Delta E$ | 3.00 | 2.95 | 2.93 | 2.94 | 2.97 | 2.95 | 3.70 | 3.69 | 2.61 | 2.58 | 2.63 | 2.60 |
| Mean $\Delta L$ | 3.96 | 4.01 | 4.03 | 4.06 | 4.01 | 4.06 | 3.95 | 3.86 | 3.97 | 2.16 | 2.15 | 2.13 |

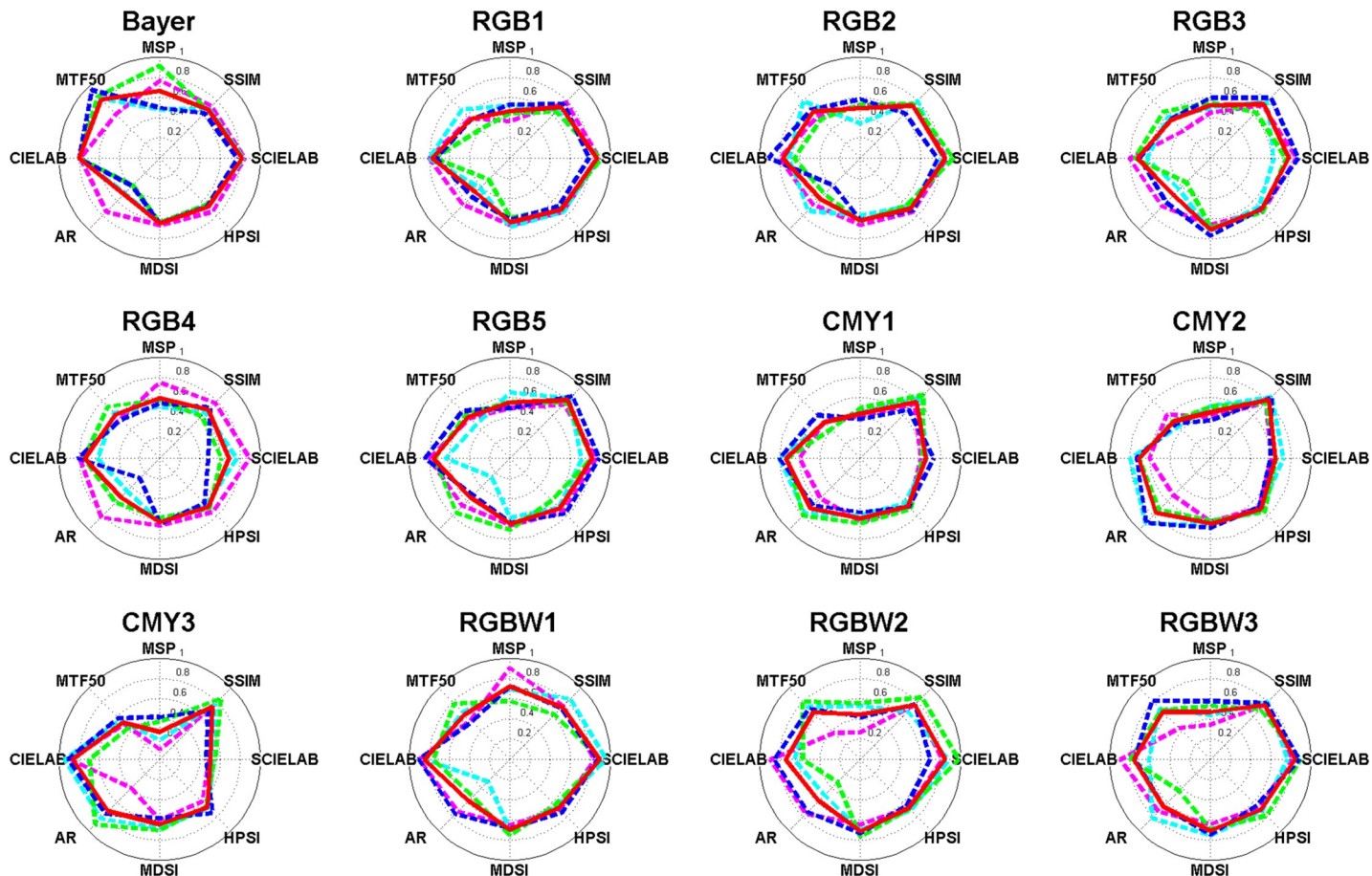

**Fig 16.** Polar coordinate visualization for test CFAs (magenta dotted line: Bilinear, cyon dotted line: laplacian, green dotted line: adaptive laplacian, blue dotted line: POCS, and red solid line: average).

should be noted that the mean AR error value of CFAs with the same elements, unlike other metrics (for example, MTF50 or MSP), is similar despite the change in structure and location of the elements. As a result, the mean AR error (0.7527) of CMY CFA is somewhat higher than that of Bayer CFA (0.6234), RGB CFA (0.6511), and RGBW CFA (0.6289). In addition, RGB4 in the RGB CFAs has the lowest mean AR error whereas CMY3 shows the highest one.

Fig 15 shows the MDSI and HaarPSI results for the test CFAs. MDSI means that the closer to 0, the higher the structural and color similarity. All test CFAs show relatively good MDSI values. The closer the GCS is to 1, the higher the structural and color similarity between the original and rendered image. Most CFAs have a lower GCS near indigo (or blue) color, which means that the similarity is lower in the area. MDSI values are similar for each CFA, however the GCS distribution is different (especially in the red and blue color regions). Based on this phenomenon, it can be deduced that MDSI value may vary according to color distribution of the input image. For each test CFAs, HaarPSI ranges 0.60 to 0.75. Total images rendered show entirely high HaarPSI value because the degree of distortion is weak compared to the original image. In perceptual similarity comparison, RGB-based CFAs show higher HaarPSI values, while CMY or RGBW-based CFAs have rather lower HaarPSI values.

Table 2 shows the results of each metric for test CFAs using different demosaicing methods. The red box represents the highest scoring CFAs in the demosaicing and metrics. RGB4 for bilinear demosaicing, RGBW1 for laplacian demosaicing, Bayer for adaptive laplacian

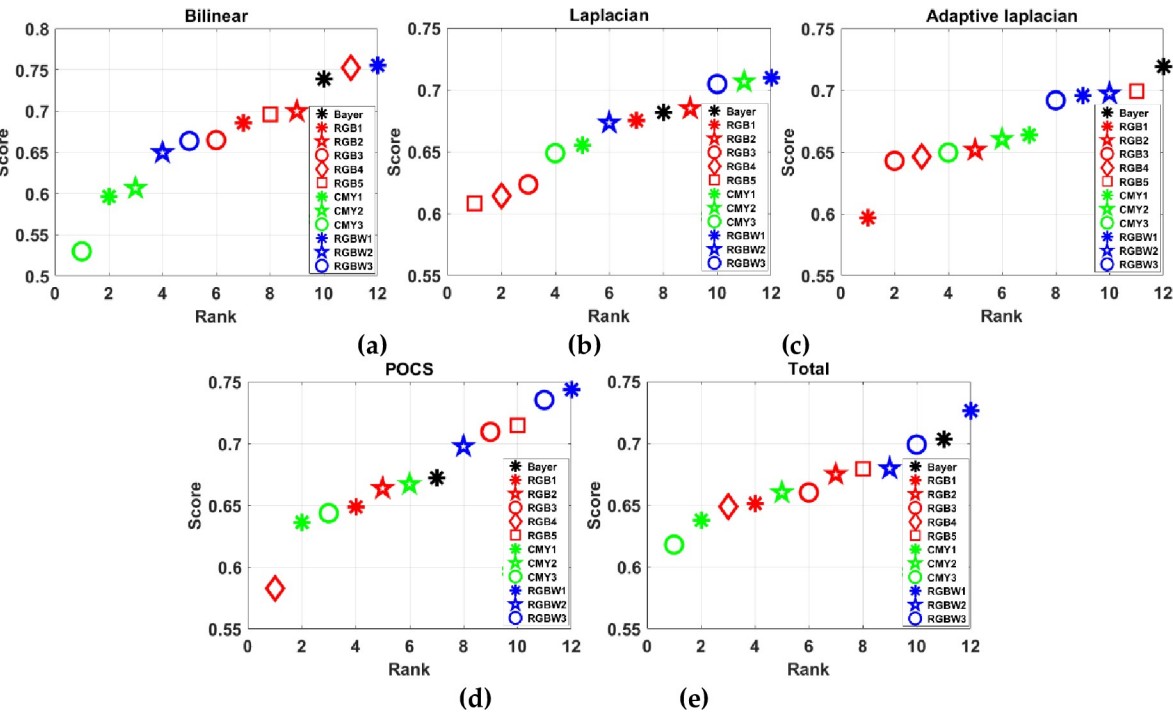

**Fig 17.** CFA ranks for (a) blinear, (b) laplacian, (c) adaptive laplacian, (d) POCS, and (e) total demosaicing methods.

demosaicing, and RGB5 for POCS demosaicing have the highest number of red box (best metric-score). Also, regardless of demosaicing methods, Bayer, RGB1~RGB5, CMY1~CMY3, and RGBW1~RGBW3 CFA have seven, two, zero, one, three, three, one, one, four, seven, two, and two red boxes respectively.

Fig 16 shows a polar coordinate visualization of the test CFAs through the proposed CFA IQ evaluation system. It can be seen that the performance of the test CFAs can be easily visualized. Table 3 shows the performance comparison of test CFAs as the average of the normalized metrics mentioned in section 2.2.9. Bayer shows the best performance for adaptive laplacian demosaicing and obtained higher metric scores of 0.0203~0.1228 compared to the other test CFAs. On the other hand, RGBW1 shows the best performance for bilinear, laplacian, and POCS demosacing and received superior metric socres of 0.0026~0.2323, 0.003~0.1014, 0.0084~0.1611 for the respective demosaicing methods. For all the demosaicing methods used in this paper, the best CFA was RGBW1 and acquired higher scores of 0.0228 to 0.1081 compared to the other test CFAs. Based on the analysis in Table 3, the CFA ranks for the respective and total demosaicing methods are shown in Fig 17. We can see that the metric score difference between the worst and the best CFA for each demosaicing method is significant.

As a result, the proposed CFA IQ evaluation system can be useful for analyzing the IQ characteristics of existing CFA structures or for evaluating the IQ when developing a CFA with a new structure. The respective metrics used in this paper are an example of analyzing a CFA. The existing or proposed metrics used in this paper evaluate quantitatively and objectively the images rendered by CFAs. In future research, we will incorporate the psychophysical (subjective) assessment factors into CFA image quality assessment, considering various experimental methods such as participants (experts and non-experts), experimental images, experimental settings such as background illumination and gamma correction of a monitor, and online or in-situ site selection.

**Table 2. Comparison results of all metrics for test CFAs according to demosaicing methods.**

| | Metrics | Bayer | RGB1 | RGB2 | RGB3 | RGB4 | RGB5 | CMY1 | CMY2 | CMY3 | RGBW1 | RGBW2 | RGBW3 |
|---|---|---|---|---|---|---|---|---|---|---|---|---|---|
| Bilinear | $\Delta E_s$ | 3.58 | 3.25 | 3.47 | 3.50 | **3.24** | 3.46 | 5.28 | 5.28 | 6.09 | 3.61 | 3.79 | 3.80 |
| | SSIM | 0.73 | **0.79** | 0.75 | 0.74 | 0.78 | 0.76 | 0.76 | 0.78 | 0.66 | 0.75 | 0.76 | 0.76 |
| | MSP | 120.00 | 68.00 | 88.00 | 50.00 | 118.00 | 85.00 | 75.00 | 74.00 | 33.00 | **138.00** | 55.00 | 65.00 |
| | MTF50 | **79.20** | 75.00 | 77.60 | 60.80 | 74.40 | 75.20 | 70.90 | 78.80 | 70.00 | 74.20 | 60.00 | 65.20 |
| | $\Delta E$ | 3.00 | 2.95 | 2.95 | 2.94 | 2.97 | 2.95 | 3.70 | 3.69 | 2.61 | **2.58** | 2.60 | 2.60 |
| | $\Delta L$ | 0.62 | 0.67 | 0.67 | 0.66 | **0.59** | 0.67 | 0.72 | 0.74 | 0.80 | 0.63 | 0.62 | 0.63 |
| | MDSI | 0.34 | 0.34 | 0.34 | 0.34 | **0.33** | 0.34 | 0.37 | 0.37 | **0.40** | 0.35 | 0.36 | 0.36 |
| | HaarPSI | **0.75** | 0.75 | 0.75 | 0.74 | 0.75 | 0.75 | 0.68 | 0.69 | 0.60 | 0.70 | 0.68 | 0.68 |
| Laplacian | $\Delta E_s$ | 3.67 | 3.38 | 3.75 | 5.10 | 4.14 | 4.50 | 4.87 | 4.41 | 5.86 | **2.79** | 3.34 | 4.13 |
| | SSIM | 0.69 | 0.67 | 0.81 | 0.80 | 0.64 | 0.84 | 0.85 | **0.87** | 0.78 | 0.85 | 0.67 | 0.81 |
| | MSP | 55 | 88 | 65 | 85 | 86 | 105 | 79 | 88 | 45 | **110** | 89 | 77 |
| | MTF50 | **98** | 84.2 | 94.40 | 77.0 | 81.52 | 67.80 | 69.8 | 65.6 | 74.00 | 83.00 | 89.00 | 87.60 |
| | $\Delta E$ | 2.93 | 2.93 | 3.31 | 3.60 | 3.65 | 3.55 | 2.94 | 2.97 | 2.39 | **2.80** | 3.52 | 3.68 |
| | $\Delta L$ | 0.75 | 0.79 | 0.63 | 0.78 | 0.77 | 0.87 | 0.61 | 0.54 | **0.59** | 0.84 | 0.75 | 0.59 |
| | MDSI | 0.36 | 0.32 | 0.43 | 0.28 | 0.37 | 0.41 | 0.43 | 0.34 | 0.32 | 0.30 | 0.30 | **0.25** |
| | HaarPSI | 0.66 | **0.75** | 0.68 | 0.68 | 0.65 | 0.69 | 0.63 | 0.72 | 0.69 | 0.66 | 0.71 | 0.72 |
| Adaptive Laplacian | $\Delta E_s$ | 3.74 | 3.14 | 3.01 | 4.23 | 5.21 | 3.79 | 5.17 | 4.96 | 5.64 | 3.18 | **2.64** | 3.21 |
| | SSIM | 0.68 | 0.65 | 0.76 | 0.64 | 0.59 | 0.79 | **0.89** | 0.78 | 0.86 | 0.63 | 0.87 | 0.70 |
| | MSP | **147** | 78 | 89 | 92 | 95 | 93 | 85 | 87 | 69 | 95 | 93 | 89 |
| | MTF50 | **104.4** | 63.00 | 74.00 | 82.6 | 88.23 | 79.3 | 63 | 72.2 | 65.40 | 92.8 | 95.2 | 85.20 |
| | $\Delta E$ | **2.95** | 3.12 | 3.51 | 3.12 | 3.14 | 3.15 | 3.24 | 3.31 | 3.34 | 3.11 | 3.83 | 2.97 |
| | $\Delta L$ | 0.81 | 0.85 | 0.75 | 0.83 | 0.69 | 0.62 | 0.60 | 0.64 | **0.55** | 0.74 | 0.84 | 0.78 |
| | MDSI | 0.36 | 0.41 | 0.37 | 0.31 | 0.41 | 0.29 | 0.35 | 0.38 | 0.30 | 0.25 | **0.23** | 0.31 |
| | HaarPSI | 0.65 | 0.71 | 0.66 | 0.75 | 0.70 | 0.59 | 0.71 | 0.76 | 0.66 | 0.64 | 0.69 | **0.79** |
| POCS | $\Delta E_s$ | 4.08 | 3.98 | 4.11 | 3.38 | 6.13 | **3.29** | 4.38 | 5.33 | 6.28 | 3.38 | 4.63 | 3.31 |
| | SSIM | 0.63 | 0.76 | 0.63 | 0.85 | 0.71 | **0.88** | 0.68 | 0.86 | 0.68 | 0.72 | 0.77 | 0.79 |
| | MSP | 70 | 89 | 96 | 98 | 91 | 85 | 71 | 69 | 75 | **115** | 75 | 95 |
| | MTF50 | **111.8** | 73.80 | 84.80 | 74.0 | 73.64 | 83.40 | 78.40 | 69.0 | 76.40 | 72.6 | 87.4 | 96.00 |
| | $\Delta E$ | 2.94 | 3.21 | 2.54 | 3.33 | 2.89 | 2.79 | 2.94 | 3.20 | **2.53** | 2.56 | 2.75 | 3.15 |
| | $\Delta L$ | 0.80 | 0.73 | 0.81 | 0.69 | 0.86 | 0.74 | 0.67 | 0.55 | 0.62 | **0.61** | 0.63 | 0.67 |
| | MDSI | 0.35 | 0.39 | 0.39 | **0.23** | 0.35 | 3.36 | 0.46 | 0.31 | 0.42 | 0.32 | 0.27 | 0.25 |
| | HaarPSI | 0.67 | 0.68 | 0.74 | 0.71 | 0.63 | **0.78** | 0.68 | 0.69 | 0.75 | 0.74 | 0.65 | 0.62 |

# 4. Conclusions

This paper presented a novel CFA IQ evaluation system that enables a comparative study of the IQM of output images rendered through various CFA patterns. Although many CFA patterns have been developed over the past few decades, it remains a challenge to design and

**Table 3. Performance comparison of test CFAs by normalized metrics.**

| Demosaic | Bayer | RGB1 | RGB2 | RGB3 | RGB4 | RGB5 | CMY1 | CMY2 | CMY3 | RGBW1 | RGBW2 | RGBW3 |
|---|---|---|---|---|---|---|---|---|---|---|---|---|
| Bilnear | 0.7391 | 0.6853 | 0.6995 | 0.6648 | 0.7526 | 0.6958 | 0.5965 | 0.6064 | 0.5299 | **0.7552** | 0.6495 | 0.6637 |
| Laplacian | 0.6820 | 0.6755 | 0.6849 | 0.6235 | 0.6145 | 0.6082 | 0.6554 | 0.7066 | 0.6488 | **0.7096** | 0.6733 | 0.7047 |
| Adaptive laplacian | **0.7195** | 0.5967 | 0.6515 | 0.6428 | 0.6463 | 0.6992 | 0.6637 | 0.6603 | 0.6495 | 0.6959 | 0.6973 | 0.6918 |
| POCS | 0.6724 | 0.6486 | 0.6637 | 0.7096 | 0.5827 | 0.7149 | 0.6361 | 0.6671 | 0.6437 | **0.7438** | 0.6978 | 0.7354 |
| Total | 0.7033 | 0.6515 | 0.6749 | 0.6602 | 0.6490 | 0.6795 | 0.6379 | 0.6601 | 0.6180 | **0.7261** | 0.6795 | 0.6989 |

analyze new CFA patterns for improving the IQ and color reproduction. The proposed CFA evaluation system includes newly devised metrics such as MSP and AR, as well as existing metrics such as CIELAB, S-CIELAB, SSIM, MTF50, MDSI, and HaarPSI, to evaluate CFA patterns and demosaicing methos from various perspectives of color accuracy, color reproduction error, AR, structural information, image contrast, moiré robustness, structual distortion, and perceptual similarity for rendered output images. To analyze the CFA IQ performance more precisely, any parameters concerning the applied metrics can be modified, or novel quantitative metrics can be added in the evaluation system.

## Supporting information

**S1 Data.**
(MAT)

**S1 Fig.**
(JPG)

**S2 Fig.**
(TIF)

**S1 File.**
(ZIP)

**S2 File.**
(ZIP)

**S3 File.**
(ZIP)

## Acknowledgments

The author is grateful to Steven Lansel, a software engineer; Oculus VR; Facebook Technologies; Munenori Fukunishi, the Division of Information Sciences, Chiba University, Japan; Prof. Brian A. Wandell, Stanford Psychology; and Joyce Farrell, the Executive Director of the Stanford Center for Image Systems Engineering, Stanford University.

## Author Contributions

**Conceptualization:** Tae Wuk Bae.

**Data curation:** Tae Wuk Bae.

**Formal analysis:** Tae Wuk Bae.

**Funding acquisition:** Tae Wuk Bae.

**Investigation:** Tae Wuk Bae.

**Methodology:** Tae Wuk Bae.

**Project administration:** Tae Wuk Bae.

**Resources:** Tae Wuk Bae.

**Software:** Tae Wuk Bae.

**Supervision:** Tae Wuk Bae.

**Validation:** Tae Wuk Bae.

**Visualization:** Tae Wuk Bae.

**Writing – original draft:** Tae Wuk Bae.

**Writing – review & editing:** Tae Wuk Bae.

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
