## [Decision Letter · Decision Letter 0]

6 Jan 2020

PONE-D-19-25883

Image Quality Metric System for Color Filter Array (CFA) Evaluation

PLOS ONE

Dear Dr. Bae,

Thank you for submitting your manuscript to PLOS ONE. After careful consideration, we feel that it has merit but does not fully meet PLOS ONE’s publication criteria as it currently stands. Therefore, we invite you to submit a revised version of the manuscript that addresses the points raised during the review process.

We would appreciate receiving your revised manuscript by Feb 20 2020 11:59PM. To enhance the reproducibility of your results, we recommend that if applicable you deposit your laboratory protocols in protocols.io, where a protocol can be assigned its own identifier (DOI) such that it can be cited independently in the future. For instructions see: http://journals.plos.org/plosone/s/submission-guidelines#loc-laboratory-protocols

We look forward to receiving your revised manuscript.

Kind regards,

Hocine Cherifi

Academic Editor

PLOS ONE

Journal Requirements:

'No'

Please provide an amended Funding Statement that declares *all* the funding or sources of support received during this specific study (whether external or internal to your organization) as detailed online in our guide for authors at http://journals.plos.org/plosone/s/submit-now Please state what role the funders took in the study.  If any authors received a salary from any of your funders, please state which authors and which funder. If the funders had no role, please state: "The funders had no role in study design, data collection and analysis, decision to publish, or preparation of the manuscript."

Reviewers' comments:

Reviewer's Responses to Questions

**Comments to the Author**

1. Is the manuscript technically sound, and do the data support the conclusions?

Reviewer #1: Partly

Reviewer #2: Partly

2. Has the statistical analysis been performed appropriately and rigorously? 

Reviewer #1: N/A

Reviewer #2: N/A

3. Have the authors made all data underlying the findings in their manuscript fully available?

Reviewer #1: No

Reviewer #2: Yes

4. Is the manuscript presented in an intelligible fashion and written in standard English?

Reviewer #1: Yes

Reviewer #2: Yes

5. Review Comments to the Author

Reviewer #1: The authors propose a method to evaluate the mosaic of CFA camera. The topic is very interesting and the proposal quite convincing, though the evaluation of the protocol and the conclusions are limited.

Hereafter are some comments:

You need to position your work vs. this article:

A no-reference metric for demosaicing artifacts that fits psycho-visual experiments

F Gasparini, F Marini, R Schettini, M Guarnera - EURASIP Journal on Advances in Signal Processing, 2012

Not clear how the mosaics, in particular complex/pseudo-random mosaics or general mosaics (SFA) will impact the relevant of the method. In particular in the case of a color transform from N bands to colour.

Random color filter arrays are better than regular ones

P Amba, J Dias, D Alleysson - Color and Imaging Conference, 2016

and

Multispectral filter arrays: Recent advances and practical implementation

PJ Lapray, X Wang, JB Thomas, P Gouton - Sensors, 2014

Neither the role of the algorithm definition:

Xin Li, Bahadir Gunturk, Lei Zhang, "Image demosaicing: a systematic survey," Proc. SPIE 6822, Visual Communications and Image Processing 2008, 68221J (28 January 2008);

and

Demosaicing of periodic and random color filter arrays by linear anisotropic diffusion

JB Thomas, I Farup - Color and Imaging Conference, 2018

I think this needs to be discussed, because the simulation and evaluation of your framework is limited to regular patterns and very basic demosaicing. It is very difficult to study mosaic alone in this case and get strong conclusion. In fact in particular Moire might be also limited by the use of better algorithm combined to each mosaic, which limit the conclusions.

That does not impact the definition of the evaluation method though.

It is not very well described what are the characteristics of the spectral data used. This needs to be described because it may impair the simulation.

Beside I encourage to revise the references: there were many online accessed documents. Some might be find in academic articles properly cited rather than a link, this must be fixed.

As well, you may provide more specific acknowledgement to what the cited person have actually done to desserve those.

English and text could be improved.

Reviewer #2: The article presents that: currently the most frequently used method of demosaicing in terms of cost and time of computation is bilinear interpolation. On what basis did it say so? There is, for example, the method of the nearest neighbour interpolation which is even faster and requires much less resources.

It would be worthwhile to see how the results obtained for different CFA systems behave for different demosaicing methods.

Consideration should be given to the statement: The pipeline typically consists of demosaicking, noise reduction, white balance, CFA interpolation, color conversion, and gamma correction for rendering the sensor data.

Is the formulation correct and is this the correct order of operations?

The proposed quality assessment system for different CFA has been reduced to a selection of several metrics used to assess the quality of a digital image where a reference image is required. The article proposes visualization in the form of polar coordinates. For different CFA systems we obtain a different distribution, however, it is the subjective observer who must assess which of the measurements is more important for him.

The authors should look at another measure of image quality assessment closely correlated with HVS. Examples of such metrics are: DSCSI, MDSI, HPSI etc…

Lee, Dohyoung, and Konstantinos N. Plataniotis. "Towards a full-reference quality assessment for color images using directional statistics." IEEE Transactions on image processing 24.11 (2015): 3950-3965.

Nafchi, Hossein Ziaei, et al. "Mean deviation similarity index: Efficient and reliable full-reference image quality evaluator." IEEE Access 4 (2016): 5579-5590.

Reisenhofer, Rafael, et al. "A Haar wavelet-based perceptual similarity index for image quality assessment." Signal Processing: Image Communication 61 (2018): 33-43.

It is also worthwhile to examine the articles:

Frackiewicz, Mariusz, and Henryk Palus. "Toward a perceptual image quality assessment of color quantized images." Tenth International Conference on Machine Vision (ICMV 2017). Vol. 10696. International Society for Optics and Photonics, 2018.

Frackiewicz, Mariusz, and Henryk Palus. "New image quality metric used for the assessment of color quantization algorithms." Ninth International Conference on Machine Vision (ICMV 2016). Vol. 10341. International Society for Optics and Photonics, 2017.

6. PLOS authors have the option to publish the peer review history of their article (what does this mean?). If published, this will include your full peer review and any attached files.

Reviewer #1: No

Reviewer #2: No

---

## [Author Response · Author response to Decision Letter 0]

20 Feb 2020

Revised text based on reviewer's advice. Please refer to the attachment files (Response to reviewer). Thank you for your valuable review.

---

## [Decision Letter · Decision Letter 1]

11 Mar 2020

PONE-D-19-25883R1

Image Quality Metric System for Color Filter Array Evaluation

PLOS ONE

Dear Dr. Bae,

Thank you for submitting your manuscript to PLOS ONE. After careful consideration, we feel that it has merit but does not fully meet PLOS ONE’s publication criteria as it currently stands. Therefore, we invite you to submit a revised version of the manuscript that addresses the points raised during the review process.

We would appreciate receiving your revised manuscript by Apr 25 2020 11:59PM. To enhance the reproducibility of your results, we recommend that if applicable you deposit your laboratory protocols in protocols.io, where a protocol can be assigned its own identifier (DOI) such that it can be cited independently in the future. For instructions see: http://journals.plos.org/plosone/s/submission-guidelines#loc-laboratory-protocols

We look forward to receiving your revised manuscript.

Kind regards,

Hocine Cherifi

Academic Editor

PLOS ONE

Reviewers' comments:

Reviewer's Responses to Questions

**Comments to the Author**

1. If the authors have adequately addressed your comments raised in a previous round of review and you feel that this manuscript is now acceptable for publication, you may indicate that here to bypass the “Comments to the Author” section, enter your conflict of interest statement in the “Confidential to Editor” section, and submit your "Accept" recommendation.

Reviewer #1: (No Response)

Reviewer #2: All comments have been addressed

2. Is the manuscript technically sound, and do the data support the conclusions?

Reviewer #1: Partly

Reviewer #2: Yes

3. Has the statistical analysis been performed appropriately and rigorously? 

Reviewer #1: Yes

Reviewer #2: No

4. Have the authors made all data underlying the findings in their manuscript fully available?

Reviewer #1: No

Reviewer #2: Yes

5. Is the manuscript presented in an intelligible fashion and written in standard English?

Reviewer #1: No

Reviewer #2: Yes

6. Review Comments to the Author

Reviewer #1: The work is improved, however there are still quite strong limitations:

2 major conceptual concerns:

1-It is not only the demosacing that seems to be evaluated, but the whole imaging pipeline, as shown in Fig 2. At least

2-Generally the writing part should be reworked and be more structured, accurate and concise.

Less major comments:

1-Do you really need to introduce all the formula for the IQMs?

2-The results should be supported by a psychophysical experiment OR why you do not do it should be very clearly stated (in this case you seem to enforce that the recent IQM correlate with perception, but then you use many metrics that do not correlate very well with perception).

3-An exercise of quantitative summary/analysis of the results would be welcome.

4-I invite you to revisit my first rounds of comments.

Minor comments:

1-Figs are not readible because too small.

2-PUPPY image and other images are not referenced and it is not sure if the simulation was in spectral or in colour only.

In general I would like this work to be published, but would appreciate it to be more compact. Beside I am not sure that Plos One is the right support for this technical development (up to the editors to take this decision though)

Reviewer #2: Are there statistically significant differences between the CFA systems tested for quality indices?

In Table 1 there are identical values for HPSI, MDSI and others for different CFA systems, maybe you need to use a different way of presenting the results?

7. PLOS authors have the option to publish the peer review history of their article (what does this mean?). If published, this will include your full peer review and any attached files.

Reviewer #1: No

Reviewer #2: No

---

## [Author Response · Author response to Decision Letter 1]

1 Apr 2020

Reviewer #1: The work is improved, however there are still quite strong limitations:

2 major conceptual concerns:

1-It is not only the demosacing that seems to be evaluated, but the whole imaging pipeline, as shown in Fig 2. At least 

Sol) The entire image pipeline, including demosaicing part in Figure 2, has been modified as shown below. And the explanation is also added.

Figure 2. Proposed CFA image-quality evaluation system.

Figure 2 shows the imaging pipeline for the proposed CFA IQ evaluation system. In the proposed system, the CFA structure and demosaicing method are changeable, and the CFA IQ evaluation results are plotted on the polar coordinates.

2-Generally the writing part should be reworked and be more structured, accurate and concise. 

Sol) The figures and writing parts of the whole paper have been revised. Duplicates have been removed from the paper, and the paper has been organized to be more concise.

Less major comments:

1-Do you really need to introduce all the formula for the IQMs ?

Sol) The formulas for IQM have changed according to the advice you have given.

(change) equations 1 to 24 -> equations 1 to 9

(deleted) The existing equations 1-4, 8-11, and 15 have been deleted.

(integrated) The equations 17-20 and 21-24 have been changed to equations 8 and 9 respectively.

2-The results should be supported by a psychophysical experiment OR why you do not do it should be very clearly stated (in this case you seem to enforce that the recent IQM correlate with perception, but then you use many metrics that do not correlate very well with perception). 

Sol) Right. Good opinion. We are considering a psychophysical (subjective) experiment. The figure below is an example of a program tool for comparing the quality of images rendered by a CFA. The tool sequentially shows a pair of images generated by test CFAs, and the experiment participants select a preferred image from the two images. Whenever images are selected, all images are sorted in order of preference by bubble sorting. Currently, we are developing a prototype of this tool, and we are contemplating participants (experts and non-experts), experimental images such as experimental images, background illumination and gamma correction of the monitor, and online or field experiment sites. Although the results of these subjective experiments are not reflected in the current paper, they will be included in future image quality metrics. Considering the current situation, the following were included at the end of section 3 (Results and Discussion).

(Line 508) The existing or proposed metrics used in this paper evaluate quantitatively and objectively the images rendered by CFAs. In future research, we will incorporate the psychophysical (subjective) assessment factors into CFA image quality assessment, considering various experimental methods such as participants (experts and non-experts), experimental images, experimental settings such as background illumination and gamma correction of a monitor, and online or in-situ site selection.

<Application for comparing images rendered by CFA>

3-An exercise of quantitative summary/analysis of the results would be welcome.

Sol) A quantitative analysis was conducted and Figure 17 was added as follows.

(Line 490) Bayer shows the best performance for adaptive laplacian demosaicing and obtained higher metric scores of 0.0203~0.1228 compared to the other test CFAs. On the other hand, RGBW1 shows the best performance for bilinear, laplacian, and POCS demosacing and received superior metric socres of 0.0026~0.2323, 0.003~0.1014, 0.0084~0.1611 for the respective demosaicing methods. For all the demosaicing methods used in this paper, the best CFA was RGBW1 and acquired higher scores of 0.0228 to 0.1081 compared to the other test CFAs. Based on the analysis in Table 3, the CFA ranks for the respective and total demosaicing methods are shown in Figure 17. We can see that the metric score difference between the worst and the best CFA for each demosaicing method is significant.

Figure 17. CFA rank for (a) Blinear, (b) Laplacian, (c) Adaptive laplacian, (d) POCS, and (e) Total demosaicing.

4-I invite you to revisit my first rounds of comments.

Sol) The revisit is as follows.

You need to position your work vs. this article:

[48] Gasparini F, Marini F, Schettini R, Guarnera M. A no-reference metric for demosaicing artifacts that fits psycho-visual experiments. EURASIP Journal on Advances in Signal Processing 2012;2012;1-15. https://doi.org/10.1186/1687-6180-2012-123.

(Line 95) Gasparini etc proposed a no-reference metric for measuring demosaicing artifacts through psycho-visual experiments [48]. Using a psycho-visual comparison test adopting a single or double stimulus method, it analyzes the subjective evaluation of the demosaicing artifacts. Then, it introduce a no-reference metric for demosaicing artifacts based on measures of blurriness, chromatic and achromatic distortions that are able to fit psycho-visual experiments. While the method focuses on a no-reference metric definition of subjective (perceptual) IQ assessment for demosaicing methods in a given CFA structure, this paper introduces a combination of proven metrics for automatic and objective IQ evaluation for CFA structures as well as demosaicing methods. 

Minor comments:

1-Figs are not readible because too small.

Sol) Figure 11, 12, 13, 14, 15, and 18 have been modified for visibility.

2-PUPPY image and other images are not referenced and it is not sure if the simulation was in spectral or in colour only.

Sol) The test input images are referenced as the following.

a) (Line 149) SLANTED-BAR (ISO 12233 resolution chart) [51] for calculating the image contrast using MTF50 

b) (Line 150) MCC [52] for measuring the color error using CIELAB

c) (Line 150) PUPPY [53] for measuring the color reproduction error (visible distortion)

[51] Williams D. Benchmarking of the ISO 12233 slanted-edge spatial frequency response plug-in. Proc. IS&T's PICS Conference. Portland, May 1998, 133-136.

[52] ColorChecker Charts. Available online:https://www.webcitation.org/671Lyp9Bu?url=http://xritephoto.com/documents/literature/en/ColorData-1p_EN.pdf (accessed on 29 March 2020).

[53] Multispectral. https://github.com/ISET/isetcam/blob/master/data/images/multispectral/StuffedAnimals_tungsten-hdrs.mat (accessed on 29 March 2020).

(Line 155) Of the test input images, PUPPY is the only multipectral scene. The sensor response of multispectral scenes is calculated, then CIE XYZ value at each pixel location is computed by the ISET camera simulator. MCC color image was created based on the Gretag-MCC [52]. The rest of the test input images are color images created by patterns generated by the algorithm.

In general I would like this work to be published, but would appreciate it to be more compact. Beside I am not sure that Plos One is the right support for this technical development (up to the editors to take this decision though) 

<I appreciate for your precious comments !!>

Reviewer #2: Are there statistically significant differences between the CFA systems tested for quality indices?

Sol) We can see that the metric score difference between the worst and the best CFA for each demosaicing method is significant. A quantitative analysis was conducted and Figure 17 was added as follows.

(Line 490) Bayer shows the best performance for adaptive laplacian demosaicing and obtained higher metric scores of 0.0203~0.1228 compared to the other test CFAs. On the other hand, RGBW1 shows the best performance for bilinear, laplacian, and POCS demosacing and received superior metric socres of 0.0026~0.2323, 0.003~0.1014, 0.0084~0.1611 for the respective demosaicing methods. For all the demosaicing methods used in this paper, the best CFA was RGBW1 and acquired higher scores of 0.0228 to 0.1081 compared to the other test CFAs. Based on the analysis in Table 3, the CFA ranks for the respective and total demosaicing methods are shown in Figure 17. We can see that the metric score difference between the worst and the best CFA for each demosaicing method is significant.

Figure 17. CFA rank for (a) Blinear, (b) Laplacian, (c) Adaptive laplacian, (d) POCS, and (e) Total demosaicing.

In Table 1 there are identical values for HPSI, MDSI and others for different CFA systems, maybe you need to use a different way of presenting the results?

Sol) The analysis was conducted as follows.

(Line 468) MDSI values are similar for each CFA, however the GCS distribution is different (especially in the red and blue color regions). Based on this phenomenon, it can be deduced that MDSI value may vary according to color distribution of the input image. For each test CFAs, HaarPSI ranges 0.60 to 0.75. Total images rendered show entirely high HaarPSI value because the degree of distortion is weak compared to the original image.

<I appreciate for your precious comments !!>

---

## [Decision Letter · Decision Letter 2]

20 Apr 2020

Image Quality Metric System for Color Filter Array Evaluation

PONE-D-19-25883R2

Dear Dr. Bae,

We are pleased to inform you that your manuscript has been judged scientifically suitable for publication and will be formally accepted for publication once it complies with all outstanding technical requirements.

With kind regards,

Hocine Cherifi

Academic Editor

PLOS ONE

Additional Editor Comments (optional):

The PLOS Data policy requires authors to make all data underlying the findings described in their manuscript fully available without restriction, with rare exception (please refer to the Data Availability Statement in the manuscript PDF file). The data should be provided as part of the manuscript or its supporting information, or deposited to a public repository. For example, in addition to summary statistics, the data points behind means, medians and variance measures should be available. If there are restrictions on publicly sharing data—e.g. participant privacy or use of data from a third party—those must be specified

Reviewers' comments:

Reviewer's Responses to Questions

**Comments to the Author**

1. If the authors have adequately addressed your comments raised in a previous round of review and you feel that this manuscript is now acceptable for publication, you may indicate that here to bypass the “Comments to the Author” section, enter your conflict of interest statement in the “Confidential to Editor” section, and submit your "Accept" recommendation.

Reviewer #1: All comments have been addressed

Reviewer #2: All comments have been addressed

2. Is the manuscript technically sound, and do the data support the conclusions?

Reviewer #1: Yes

Reviewer #2: Yes

3. Has the statistical analysis been performed appropriately and rigorously? 

Reviewer #1: Yes

Reviewer #2: Yes

4. Have the authors made all data underlying the findings in their manuscript fully available?

Reviewer #1: No

Reviewer #2: Yes

5. Is the manuscript presented in an intelligible fashion and written in standard English?

Reviewer #1: Yes

Reviewer #2: Yes

6. Review Comments to the Author

Reviewer #1: Thanks for answering my comments.

The article is OK for publication. The experiment is described clearly, which was one of my major concern.

Reviewer #2: The author took into account previous comments and advice. I don't have any additional suggestions for work.

7. PLOS authors have the option to publish the peer review history of their article (what does this mean?). If published, this will include your full peer review and any attached files.

Reviewer #1: No

Reviewer #2: No

---

## [Editor Report · Acceptance letter]

23 Apr 2020

PONE-D-19-25883R2 

Image-Quality Metric System for Color Filter Array Evaluation 

Dear Dr. Bae:

I am pleased to inform you that your manuscript has been deemed suitable for publication in PLOS ONE. Congratulations! Your manuscript is now with our production department. 

With kind regards,

on behalf of

Professor Hocine Cherifi 

Academic Editor

PLOS ONE